# Loss of PRC2 subunits primes lineage choice during exit of pluripotency

Chet H. Loh[1], Siebe van Genesen[1], Matteo Perino [1,2], Magnus R. Bark[1] & Gert Jan C. Veenstra [1✉]

Polycomb Repressive Complex 2 (PRC2) is crucial for the coordinated expression of genes during early embryonic development, catalyzing histone H3 lysine 27 trimethylation. Two distinct PRC2 complexes, PRC2.1 and PRC2.2, contain respectively MTF2 and JARID2 in embryonic stem cells (ESCs). In this study, we explored their roles in lineage specification and commitment, using single-cell transcriptomics and mouse embryoid bodies derived from *Mtf2* and *Jarid2* null ESCs. We observe that the loss of *Mtf2* results in enhanced and faster differentiation towards cell fates from all germ layers, while the *Jarid2* null cells are predominantly directed towards early differentiating precursors, with reduced efficiency towards mesendodermal lineages. These effects are caused by derepression of developmental regulators that are poised for activation in pluripotent cells and gain H3K4me3 at their promoters in the absence of PRC2 repression. Upon lineage commitment, the differentiation trajectories are relatively similar to those of wild-type cells. Together, our results uncover a major role for MTF2-containing PRC2.1 in balancing poised lineage-specific gene activation, whereas the contribution of JARID2-containing PRC2 is more selective in nature compared to MTF2. These data explain how PRC2 imposes thresholds for lineage choice during the exit of pluripotency.

[1] Department of Molecular Developmental Biology, Faculty of Science, Radboud Institute for Molecular Life Sciences, Radboud University, Nijmegen, The Netherlands. [2]Present address: Genome Biology Unit, European Molecular Biology Laboratory (EMBL), Heidelberg, Germany. ✉email: g.veenstra@science.ru.nl

During early stages of mammalian development, the epiblast receives both inductive and repressive cues to precisely regulate the exit of pluripotency and the onset of differentiation. These cues, especially signaling morphogens, pattern the embryo, establish the body axes, and specify lineages through dynamic and temporal gradients, eventually producing different cell types from the distinct embryonic germ layers (endoderm, mesoderm, and ectoderm)[1–3]. To do that efficiently, pluripotent inner cell mass and embryonic stem cells (ESCs) derived from them, employ a multitude of highly conserved genetic mechanisms to repress or activate the genes in a spatially and temporally controlled fashion.

Polycomb Repressive Complex 2 (PRC2) is one of the key transcriptional repressors in ESCs. It methylates lysine 27 of histone H3 (H3K27me3), marking the chromatin for compaction, and repressing target genes together with polycomb repressive complex 1 (PRC1)[4–8]. In ESCs, specific lineage and differentiation genes like *Brachyury*, *Otx2*, and *Gata1/2* are repressed by PRC2 during pluripotency[9–12]. In addition, PRC2 represses ectopic expression of lineage-specific genes, which is thought to stabilize lineage commitment. The core components of PRC2 include the catalytic proteins EZH1/2, EED, SUZ12, and RBBP7/4[13,14]. In recent years, a number of accessory subunits have been found at sub-stoichiometric levels[15–19]. This led to a classification of two distinct PRC2 subcomplexes. PRC2.1 consists of the core together with one of the Polycomblike proteins (PHF1, MTF2, or PHF19) and EPOP, whereas PRC2.2 contains the core subunits with JARID2 and AEBP2. MTF2 recruits PRC2.1 to unmethylated CpG-rich DNA[15,16]. Its significance in mESCs was underscored by the relatively strong genome-wide reduction of PRC2 binding and H3K27me3 enrichment upon the loss of *Mtf2*[15,20–22]. In PRC2.2, JARID2 can bind DNA and also nucleosomes through recognition of ubiquitylated histone H2A (H2Aub119), a modification catalyzed by polycomb repressive complex 1 (PRC1)[23,24]. In addition, RNA is also important for the recruitment of PRC2. The association with RNA inhibits its binding to chromatin at non-target locations, thereby contributing to the proper targeting of PRC2 to developmental control genes[25,26].

While much work has been done to unravel the molecular mechanisms of PRC2 recruitment in embryonic stem cells, little is known about how the PRC2 subcomplexes affect the exit of pluripotency. In mouse ESCs, the PRC2 core subunit genes *Suz12*, *Eed*, and *Ezh2* are not required for naïve pluripotency, but they are required for the maintenance of pluripotency in the primed state[27] and for specification toward early precursors such as the primitive endoderm[28]. This presents a question of how PRC2 is regulating the exit of pluripotency, and more specifically, how it affects lineage choice. Furthermore, *Mtf2* and *Jarid2* mutants further revealed the requirement for accessory subunits during embryonic development, with mutants experiencing embryonic lethality (before E15.5 for *Mtf2* and E18.5 for *Jarid2* mutants)[29,30].

In this study, we explored how PRC2.1 and PRC2.2 mutant mESCs behaved during differentiation. Using single-cell transcriptomic analyses of mouse embryoid bodies (EBs), we observe that the *Mtf2* mutant cells differentiated faster into all germ layers, while the *Jarid2* mutants are delayed and predominantly give rise to early differentiating precursors, at the expense of mesendodermal cells. Intriguingly, we find that MTF2 represses key lineage-specific transcription factors and signaling genes which are inherently poised for activation in wild-type undifferentiated cells. With the loss of MTF2, their transcript levels and promoter H3K4me3 modifications are increased, whereas H3K27me3 is decreased, allowing for rapid induction of lineage specification genes. Loss of JARID2 derepresses a partially different and functionally more selective set of genes. As shown in directed differentiation experiments, this leads to a stronger

induction of lineage-specific gene expression in mesoderm and endoderm progenitors, while still allowing alternative lineages to be repressed.

Together, our results outline a critical role of PRC2.1 (MTF2) in controlling the threshold of lineage gene activation by maintaining repression on key lineage transcription and signaling factors, and therefore modulating the state of promoter bivalency. Furthermore, the single-cell resolution of EB differentiation reveals differences in lineage potency upon the loss of PRC2.1 and PRC2.2, which is linked to their effects on H3K27 methylation and derepression of specific genes, uncovering their roles during the exit of pluripotency.

## Results

**Embryoid bodies of PRC2 mutant ESCs differentiate to cell types across all germ layers.** Previously, we and others have observed differences in core PRC2 recruitment between the loss of MTF2 (PRC2.1 mutant) or JARID2 (PRC2.2 mutant) in mESCs[15,22,31]. This potentially affects the repression of key lineage transcription factor genes, for example *Otx2*, where EZH2 binding was dramatically reduced in the *Mtf2* null compared to the *Jarid2* null or wild-type cells (Fig. 1a).

We wondered how such a loss of PRC2 subunits affects the exit of pluripotency and early germ layer differentiation. To untangle the heterogeneity and track the differentiation lineage potential of these mutants over a broad range of cell fates, we performed mouse embryoid body (EB) differentiation (Fig. 1b) and captured the transcriptomes of over 5400 single cells, encompassing different genetic backgrounds (wild-type, *Mtf2* null, *Jarid2* null, and *Eed* null cells) and time points (day 0–10) during differentiation (Fig. 1c, d). Phenotypically, we did not observe any stark differences in shapes and sizes of differentiated EBs between the different genetic backgrounds (Fig. 1b). After stringent quality checks for outliers and dropouts (cells with little or no mRNA recovered), the remaining cells (4949) were normalized for differences in sequencing depth and batch variation (see the "Methods" section), clustered using the Louvain method for community detection, and projected in a two-dimensional space using the Uniform Manifold Approximation and Projection (UMAP) method for dimension reduction (Fig. 1c–e).

Clear differences were observed between the different Polycomb mutants during differentiation, particularly between the *Eed* (core PRC2 subunit) null cells and the rest (Fig. 1c). To examine the cellular phenotypes of the mutants and their matched wild-type cells in more detail, we performed bulk RNAseq on EB differentiation (Supplemental Fig. 1). During differentiation strong differences are observed between the lines. The undifferentiated *Eed* null cells show a large number of strongly upregulated genes, which tend to be less upregulated in *Jarid2* null cells, whereas *Mtf2* null cells show relatively strong deregulation of a partially overlapping set of genes (Supplemental Fig. 1a). We have used this data to compare with the single-cell transcriptomic profiles and characterize their differentiation characteristics (see below).

The two-dimensional UMAP representation of the single-cell data generally reproduces the temporal order of differentiation, with the early and late time points relatively more to the top and to the bottom of the graph, respectively (Fig. 1d). The pluripotency markers *Esrrb* and *Nanog* were expressed in cell clusters corresponding to early time points, whereas lineage-specific markers were expressed at later time points (Fig. 1f–i; Supplemental Fig. 2c–f). Generally, the undifferentiated ES cells of different genotypes occupy distinct clusters, but some of these cells converge into mixed genotype clusters upon differentiation (Fig. 1c, Supplemental Fig. 2a, b). This is most dramatically observed for *Eed* null cells, which cluster quite differently from

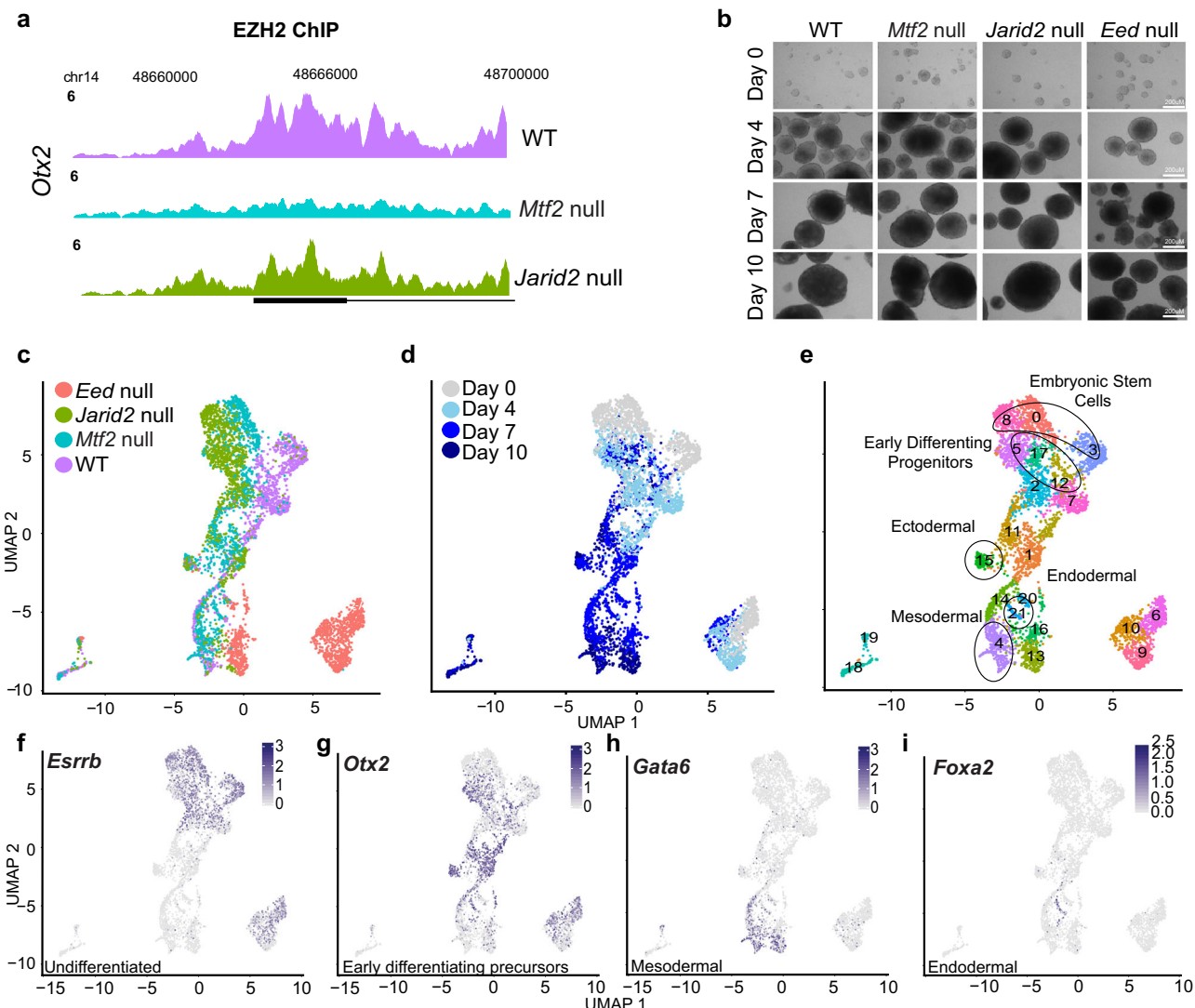

**Fig. 1 Embryoid bodies of PRC2 mutant ESCs yield a multitude of cell types across different germ layers. a** EZH2 ChIP peak profiles of gene *Otx2* for WT, Mtf2 null, and *Jarid2* null cells. Peaks were RPKM normalized and scaled across backgrounds. **b** Phase-contrast of Embryoid Bodies from different background and time point of differentiation. Each differentiation time point and background were performed in replicates. **c, d** Single-cell UMAPs of 4949 cells over all backgrounds and time points. **e** Single-cell UMAP of cells colored by predicted clusters. **f–i** Feature maps of selected lineage genes, color intensity based on normalized expression of individual gene.

the rest of the cells. Clusters 9, 10, 13, and 16, mainly *Eed* null cells, appear to be restricted in differentiation (Fig. 1e; bottom right region of UMAP) and do not share a similar path compared to the WT cells and other PRC2 subunit mutants in both early and late stage differentiation. Compared to these cells, the transcriptomes of *Mtf2* null and *Jarid2* null ES cells exhibit smaller differences with WT ES cells, but still form genotype-specific clusters when not differentiated (clusters 3, 8, and 0 for, respectively, wild type, *Mtf2* null, and *Jarid2* null cells). In the *Mtf2* null, *Jarid2* null, and WT EBs, the cells pass through largely genotype-specific clusters with early differentiation intermediates (respectively, cluster 12, 5, and 17), after which they tend to converge into mixed genotype differentiated clusters (Fig. 1c, d, Supplemental Fig. 2a).

To assist a systematic analysis of differentiation potential in relation to the loss of PRC2 subunits, we annotated each of the 22 clusters in our dataset using Anatomy Ontology[32] and used this in combination with matches to cell types in the recently published Mouse Cell Atlas[33] ("Methods"; Supplemental Fig. 3a, b). We then grouped cell clusters into the germ layers for

downstream analyses (Supplemental Fig. 3c). The EBs produce a rich diversity of cell lineages of all three germ layers. For example, cluster 4 consisted of cells resembling the lateral mesoderm lineage (e.g., heart and pericardium, mesenchyme, hematopoietic progenitors), expressing high levels of cardiac markers such as *Gata6*, *Tbx20*, and *Isl1* (Fig. 1e, h, Supplemental Figs. 2e, 3a). Clusters 15 and 1 were characterized as ectodermal cells (e.g., neural ectoderm, retina) enriched for *Otx2* expression (Fig. 1e, g, Supplemental 3a–c). Cluster 21 exhibits endoderm-specific gene expression, including the expression of genes such as *Foxa2* and *Sox17* (Fig. 1e, i, Supplemental Figs. 2f, 3a). Interestingly, many of the clusters appeared to be predominantly enriched for cells from specific PRC2 mutants (Fig. 1c, e, Supplemental Fig. 3d). This raised the question to what extent a bias in differentiation can be caused by differences between the PRC2 mutants starting from the undifferentiated state.

**Altered patterns of lineage commitment in PRC2 mutant embryoid bodies.** We first analyzed our bulk RNA-sequencing

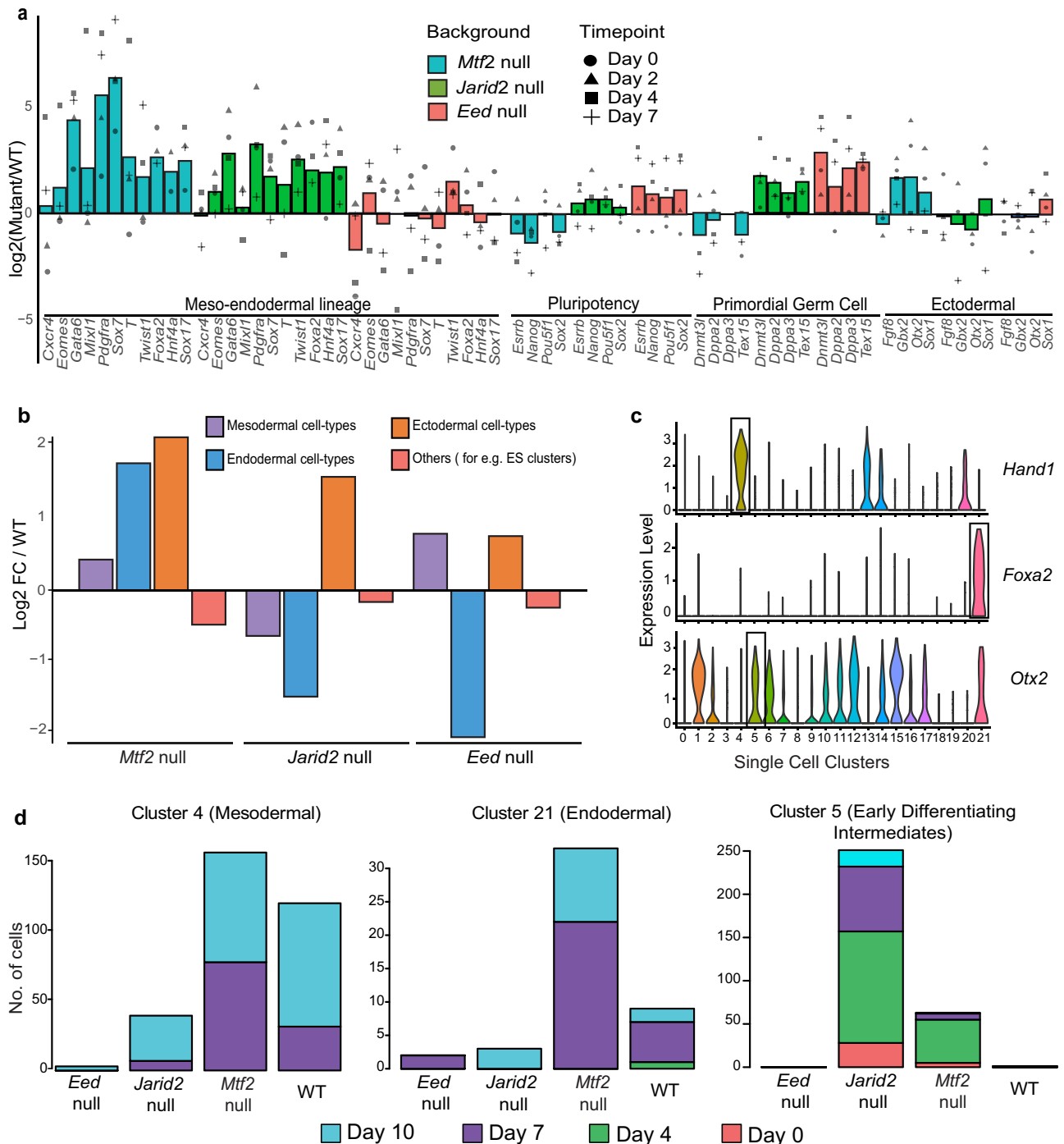

**Fig. 2 Altered patterns of lineage commitment in PRC2 mutant embryoid bodies. a** Bulk RNA-seq expression of upregulated key lineage markers in different mutant lines over their respective wild types. Bars show average values of all time points, plotted separately by shape. **b** Barplot showing enrichment of different germ layer cell types from genetic mutants compared against WT, derived from single-cell RNA-seq clusters. **c** Violin plots depicting expression of key lineage transcription factors across single-cell clusters. **d** Barplots of respective lineage clusters showing the number of cells from each time point in each single-cell cluster for respective backgrounds.

data of the Polycomb mutants and their genetically matched wild-type EBs during differentiation by calculating the fold differences of selected lineage markers between mutants and WT (Fig. 2a). We found that the *Mtf2* null cells displayed a strong differentiation propensity toward all three germ layers, with elevated expression of key mesendodermal markers such as *Mixl*1, *Eomes*, and *Pdgfra* compared to the other mutants (Fig. 2a, Supplemental Fig. 1e–g). Accordingly, the expression of pluripotent and PGC markers was relatively lower in these cells compared to *Jarid2* and

*Eed* null cells. *Jarid2* null EBs also showed increased expression of the mesendodermal lineage genes, while also showing increased of pluripotent and primordial germ cell (PGC) markers (Fig. 2a, Supplemental Fig. 1h). For *Eed* null cells, we observed PGC and reproductive organ expression signatures (Fig. 2a, Supplemental Fig. 1b–k), concordant with our single-cell analysis (*Dnmt3l*, *Dppa2*, and *Dppa3*; Supplemental Fig. 3f).

To relate these findings to our single-cell clusters, we calculated the ratios of the total number of cells from each germ layer over

the total number of cells from each genetic background (Fig. 2b, Supplemental Fig. 3c, d, Supplemental Table 1). The biological annotations of the clusters were based on their Anatomy Ontology terms and key lineage marker gene expression. Concordant with the marker gene analysis of our bulk RNA-seq data (Fig. 2a), cells with a loss of *Mtf2* were overrepresented in all three germ layers, at the expense of clusters with mixed germ layer annotations (Fig. 2b, Supplemental Fig. 3c, d) and early differentiating precursors (Supplemental Table 1). *Jarid2* null cells appear to generate more ectodermal cells at the expense of mesodermal or endodermal lineages (Fig. 2b), mainly on account of the *Otx2*-expressing neuroectodermal cluster 15, in which both *Jarid2* null and *Mtf2* null cells were overrepresented (Supplemental Fig. 3d). The bulk RNA-seq data shows that *Otx2* is more abundantly expressed in *Mtf2* null cells compared to WT cells, but this is not observed in *Jarid2* null cells (Fig. 2a). *Jarid2* null cells were also overrepresented in clusters 5 and 17, which represent early differentiation intermediates that also express *Otx2* (Fig. 1e, g). This combination of *Otx2* and pluripotency gene expression, which is also observed in a corresponding cluster of predominantly wild-type cells (cluster 12), may correspond to Rosette-stage pluripotency between naïve and primed states[34,35]. *Eed* null cells were enriched in cell types from the mesodermal and ectodermal but not endodermal lineages (Fig. 2b), in line with marker gene analysis in the bulk expression data (Fig. 2a). Clusters 9 and 10, which are predominantly *Eed* null cells show PGC and reproductive organ expression signatures (*Dnmt3l*, *Dppa2*, and *Dppa3*; Supplemental Fig. 3f), concordant with PGC marker gene expression in our bulk RNA-seq analyses (Fig. 2a, Supplemental Fig. 1c, d). *Eed* null cells produce mesodermal and ectodermal lineages at lower levels compared to *Mtf2* null cells.

**Mtf2 null cells progress faster through mid-differentiation than WT and Jarid2 null cells.** To understand the differentiation characteristics and their relation to time, we examined several clusters in our single-cell data in more detail (Fig. 2c, d). Because *Mtf2* and *Jarid2* null cells were more comparable in their undifferentiated state and their global lineage trajectories, we focused this analysis on these lines. Cluster 5, one of the clusters with early differentiation intermediates that is enriched for *Jarid2* null cells, contained cells from both late (day 7 and 10) and earlier time points (Fig. 2d, cf. Fig. 1d). Likewise, other clusters with early differentiation intermediates also contain cells from days 7–10. These clusters are enriched for *Jarid2* null cells (clusters 5, 17) or WT cells (clusters 12 and 7), but not *Mtf2* null cells (Fig. 2d, Supplemental Fig. 3e). This suggests that *Jarid2* null cells, and to some extent WT cells, are slower in their differentiation compared to *Mtf2* null cells. Indeed, clusters 4 and 21, which contain cells that are more advanced in their differentiation in the mesodermal and endodermal lineages, respectively, contain more *Mtf2* null cells compared to WT and *Jarid2* null cells. This difference is most pronounced at day 7 already, suggesting a rather efficient differentiation of *Mtf2* null cells.

To observe the process of lineage commitment in more detail, we performed an unsupervised trajectory analysis combining wild type, *Mtf2* null, and *Jarid2* null cells. These trajectories provide a common framework to compare the differentiation characteristics of cells with different mutations. Individual cells were ordered based on gene expression differences and plotted as a function of pseudotime (Fig. 3a, Supplemental Fig. 4a–d), which approximates the progress of single cells during a continuous process such as cellular differentiation.

By overlaying the experimental time points (day 0, 4, 7, and 10) onto the pseudotime plot, we observed that the ordering of cells

in the differentiation trajectory reliably recapitulated the real time of the samples (Fig. 3a, b, Supplemental Fig. 4). We observed that *Mtf2* null cells were more concentrated toward the later pseudo-times as compared to the *Jarid2* null cells (Supplemental Fig. 4b–d). To quantify the speed with which the cells differentiated, we calculated the differences in pseudotime (which is based on gene expression, representing the 'distance' of differentiation) over real-time intervals. Strikingly, we found that the *Mtf2* null cells differentiated at a faster rate, especially between days 4 and 7 of differentiation (Fig. 3c, d). We also found that the *Jarid2* null cells, which were severely delayed by day 7, partially catch up between days 7 and 10 (Fig. 3c). Finally, we analyzed the differentiation speed of Eed null cells along the rest of the cells, and found that they also differentiated at a faster rate from day 4 onward (Supplemental Fig. 5e, f). Importantly, the inclusion of the *Eed* null cells did not alter the relative ordering of *Mtf2* null, *Jarid2* null, and WT cells or their apparent differences in differentiation kinetics (Supplemental Fig. 5f).

We related these differences in trajectory speed to marker gene expression. *T (Brachyury)* was already expressed in some of the day 4 *Mtf2* null cells, as compared to day 7 for wild-type cells (Wilcoxon signed-rank test $p$ value $<4 \times 10^{-3}$; Fig. 3f). The key endodermal marker *Foxa2* was predominantly expressed in the *Mtf2* null cells (Fig. 3g; Wilcoxon signed-rank test $p$ value $<3 \times 10^{-7}$). Relatively more *Jarid2* null cells retained expression of the pluripotent marker *Klf4* at early time points (Fig. 3e), and more of them gained neuroectodermal *Pax6* expression over time (Fig. 3h). Similarly, expression of key primitive streak markers is later and slower in *Eed* null cells, but expression of PGC markers, such as *Dppa3* is much earlier (Supplemental Fig. 5g, h).

We verified these findings with marker gene expression in our bulk RNA-seq data. When compared to their genetically matched backgrounds, *Mtf2* null cells consistently showed a faster and higher expression of mesendoderm and ectodermal markers such as *T*, *Gata6*, and *Gbx2* (Supplemental Fig. 5i). *Jarid2* null cells displayed a slower downregulation of pluripotency genes such as *Nanog* and *Esrrb* and increased expression of PGC markers, whereas mesendodermal genes showed both up- and down-regulation (Supplemental Figs. 5i and 2a). *Eed* null cells demonstrated a pronounced differentiation toward the PGC state, but lower germ layer differentiation compared to both *Mtf2* and *Jarid2* null cells, which is consistent with the single-cell analyses.

In summary, our single-cell transcriptome analyses of the different PRC2 subunit mutants revealed differences in speed and direction of differentiation into germ layers precursors. Next, we investigated the gene-regulatory effects that could explain these apparent differences in differentiation.

**Derepression of PRC2 targets in Mtf2 and Jarid2 null ES cells.** We explored the gene expression differences between undifferentiated *Mtf2* and *Jarid2* null cells in more depth using our bulk RNA sequencing data (Fig. 4a, Supplemental Fig. 6a; cf. Supplemental Fig. 1). The majority of differentially expressed genes between *Mtf2* null and wild-type cells was upregulated (529 of 559 genes, log2 fold change of >2, adjusted $p$-value <0.001; Fig. 4a, b). We identified 860 genes as differentially expressed in Jarid2 null cells, 314 genes of which were upregulated (Fig. 4b).

Because PRC2 is a repressor, we overlapped the upregulated genes from mutants with a list of PRC2-bound target genes (Supplemental Data 1) which we defined by EZH2 ChIP-sequencing[15]. The upregulated genes in *Mtf2* null cells overlapped significantly with EZH2-bound genes (242 genes, hypergeometric $p$ value $<2.8 \times 10^{-62}$), whereas the downregulated genes did not show a significant overlap (Fig. 4b). We detected relatively few upregulated genes in *Jarid2* null cells that also recruited EZH2

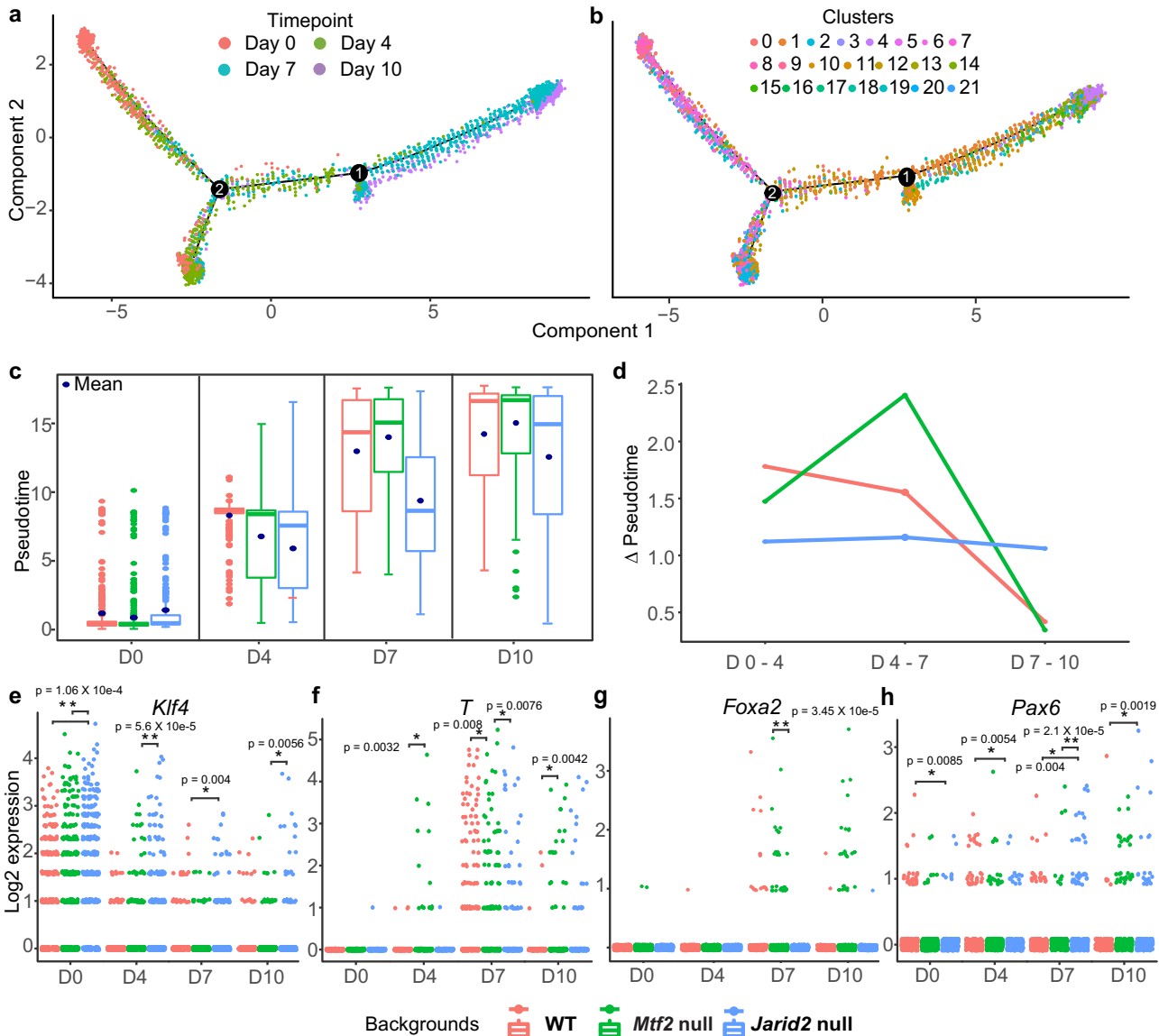

**Fig. 3 Mtf2 null cells progress faster through mid-differentiation than WT and Jarid2 null cells. a** Trajectory plot of all cells ordered along pseudotime. Two branching points were shown (numbered) along the plot. **b** Same trajectory plot as (**a**) but overlaid with cluster identity of individual cell. **c** Boxplots for each time point showing the pseudotime values of cells from different genetic backgrounds. No. of cells for each time point and background can be found in Supplemental Table 1. Dark blue dots represent the mean. Whisker ends of boxplot represent the maximum (top) and minimum values, respectively. Top and bottom of boxplots represent 75th and 25th percentile values, respectively, and finally, median values are shown as colored lines within the boxplots. **d** Delta pseudotime plot, showing the distance (change in mean pseudotime value) per unit of time (days) between different genetic backgrounds. **e–h** Log2 expression values for selected lineage genes at different time points; comparing between different genetic backgrounds. Each dot represents a single cell. Significant differences (Wilcoxon signed-rank test, two-sided alternative hypothesis testing) are indicated: *$p$-value < 0.01, **$p$-value <$1 \times 10^{-4}$. No. of cells for each time point and background can be found in Supplemental Table 1.

(58 genes, not significant; Fig. 4b). There is a small overlap between the EZH2-bound genes that are upregulated in the absence of MTF2 and JARID2 (22 genes; Supplemental Fig. 6b). We examined the levels of EZH2 recruitment of the genes upregulated in *Mtf2* and *Jarid2* null cells, and observed a reduction of EZH2 peak signals in the promoter regions in *Mtf2* null cells, whereas EZH2 recruitment to these genes was considerably less affected in *Jarid2* null cells (Fig. 4c). The upregulated genes in *Jarid2* null cells are associated with specific gene ontology terms related to early embryonic development, particularly in forebrain development and axon guidance (Fig. 4d). The 242 upregulated Polycomb target genes in *Mtf2* null cells were enriched for these and additional sets of genes,

including heart morphogenesis, gastrulation, and endoderm formation (Fig. 4d), terms that correspond to cell clusters with an overrepresentation of *Mtf2* null cells in the EB differentiation experiments.

These analyses underscore the importance of *Mtf2* relative to that of *Jarid2* in PRC2-dependent transcriptional repression of developmental genes. The loss of *Jarid2* leads to a smaller reduction of EZH2 binding and a much less profound derepression of EZH2-bound genes compared to the loss of *Mtf2*. To further understand how the upregulated PRC2 target genes in the *Mtf2* null cells affect lineage-specific gene expression, we explored the profiles of permissive H3K4me3 and repressive H3K27me3 marks at the MTF2-dependent PRC2 target genes in mESCs.

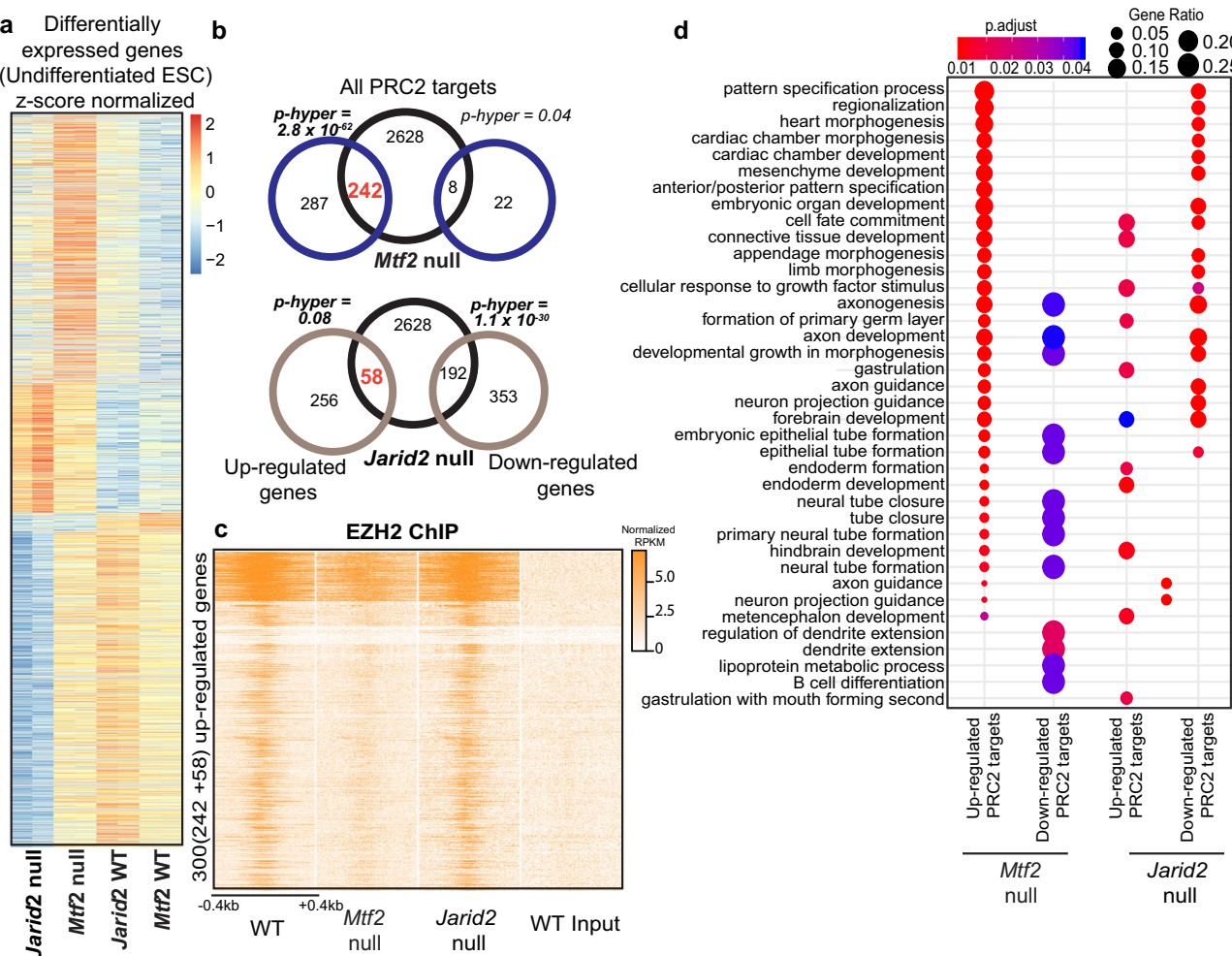

**Fig. 4 Derepression of PRC2 targets in *Mtf2* and *Jarid2* null ES cells. a** Heatmap of differentially expressed genes comparing different mutants against WT in undifferentiated mESCs. Z-score normalized counts (row scaling) are shown in heatmap. **b** Venn diagram depicting overlap of a number of genes which were up/downregulated in *Mtf2* null and *Jarid2* null cells, with PRC2 targets (as determined from EZH2 ChIP targets; next panel). Significance of overlap calculated using hyper-geometrical test. **c** Heatmap of EZH2 binding peaks (RPKM normalized) for upregulated genes (from *Mtf2* and *Jarid2* null ESCs) for different genetic backgrounds. Heatmap depicts a window of $+/-400$ bp from the transcription start site (TSS) of the genes. **d** Dot plot showing the enriched Gene Ontology terms for the differentially expressed genes in different genetic backgrounds.

**Key PRC2-repressed lineage transcription factors are poised for activation.** Lineage commitment is associated with activation of genes specific to that lineage and repressing genes of other lineages. We, therefore, hypothesized that the primed exit of pluripotency in *Mtf2* null cells is linked to the derepression of differentiation genes. First, we analyzed the levels of repressive H3K27me3 mark at key lineage transcription factor loci like *Sox11, Foxa2,* and *Gata6* in both wild-type and *Mtf2* null cells (Fig. 5a). The levels of H3K27me3 were reduced upon the loss of *Mtf2*, as expected. These levels were comparable between all the EZH2-bound genes and the subset of *Mtf2* upregulated genes, both in WT (equally high H3K27me3) and *Mtf2* null cells (equally low H3K27me3; Fig. 5b). Similarly, binding of EZH2 was comparable at all PRC2 genes compared to the subset derepressed in the absence of MTF2, whereas EZH2 binding was marginally lower at genes derepressed in the absence of JARID2 (Fig. 5b, Supplemental Fig. 7a, b). We noticed increased H3K4me3 enrichment in *Mtf2* null cells, which is observed for all PRC2 targets and the subset of *Mtf2* null-derepressed PRC2 targets (Kolmogorov–Smirnov *p*-values $<2.2 \times 10^{-16}$ and $2.1 \times 10^{-6}$, respectively). *Jarid2* null-upregulated genes have significantly higher levels of H3K4me3 when compared against all EZH2-

bound genes, even in WT cells (Kolmogorov–Smirnov *p*-value $<0.05$). Previously we found that MTF2 and JARID2 mutually stabilize their binding, which in part, happens through EED binding to H3K27me3[15,36]. We noticed that overall PRC2 binding (EZH2) is rather similar between gene sets and that the mutual destabilization of MTF2 and JARID2 binding is comparable for the 242 genes *Mtf2* null-derepressed genes compared to all PRC2 targets (Fig. 5b, Supplemental Fig. 7a, b). Similarly, the predicted DNA shape characteristics associated with MTF2-bound sequences[15] are indistinguishable (Supplemental Fig. 7c). As co-occurring H3K4me3 and H3K27me3 modifications most likely represent a bistable regulatory state[37], these observations suggest that the derepressed genes were bivalent in nature and poised for transcriptional activation (Fig. 5a, b, Supplemental Fig. 7a). This raised the question how specific transcriptional activators play a role in transcriptional derepression in the absence of MTF2.

Therefore, we analyzed transcription factor motifs to uncover gene-regulatory differences in *Mtf2* null cells versus all PRC2 targets. We regressed the presence of motifs in the promoters of all differentially expressed genes in *Mtf2* null and *Jarid2* null cells against the variance in gene expression between these lines and

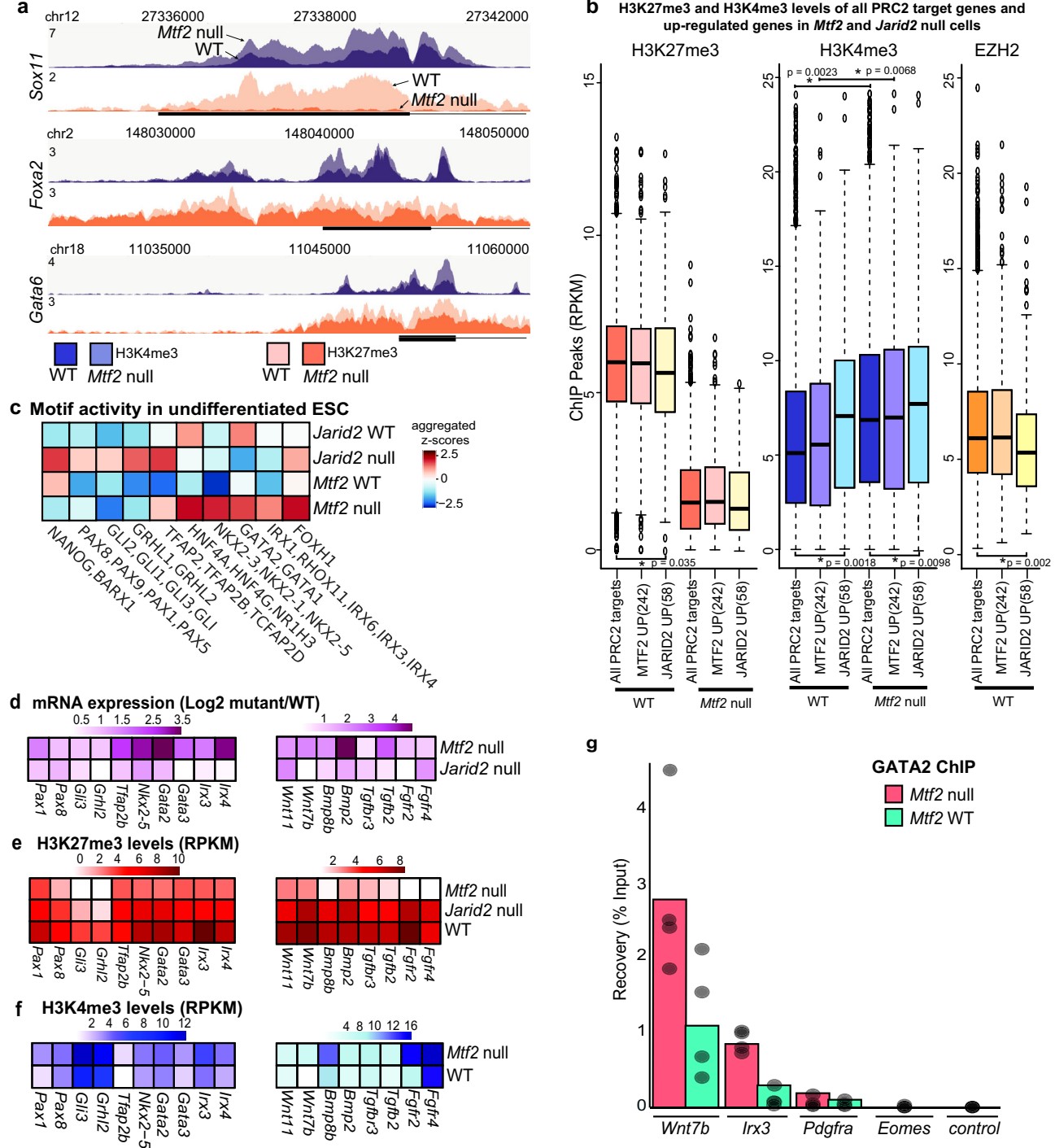

**Fig. 5 Key PRC2-repressed lineage transcription factors are poised for activation. a** ChIP peak profiles of histone mark H3K27me3 and H3K4me3 for selected upregulated (*Mtf2* null) lineage transcription factors for WT and *Mtf2* null cells at pluripotent stage. ChIP profiles were RPKM normalized and scaled between two merged profiles per histone ChIP. **b** Boxplots depicting the RPKM values of promoter (as defined by $+/-500$ bp from TSS) H3K27me3, H3K4me3, and EZH2 for all PRC2 targets and the upregulated genes in *Mtf2* null and *Jarid2* null cells. Asterisk(*) represents a Two-sample Kolmogorov–Smirnov test *p*-value <0.05. $n = 2878$ for all PRC2 targets, $n = 242$ for upregulated *Mtf2* null genes and $n = 58$ for upregulated *Jarid2* null genes. Whisker ends of boxplot represent the maximum (top) and minimum values, respectively. Top and bottom of boxplots represent 75th and 25th percentile values, respectively, and finally, median values are shown as colored lines within the boxplots. **c** Transcription factor motif activity that explains part of the variance in transcript levels, based on motifs in the promoters (as defined by $+/-500$ bp from TSS) of all upregulated PRC2-bound genes ("Methods"; upregulated genes in all four cell lines, cf. Fig. 4b). Motifs are shown in aggregated z-scores. **d**–**f** Heatmaps showing the mRNA fold change, H3K27me3 and H3K4me3 levels for a set of transcription factors (left, identified in panel **c**) and signaling factors (right, identified from *Mtf2* and *Jarid2* null DEGs list). Genes shown are all PRC2 targets. **g** Barplot depicting the GATA2 ChIP recovery relative to input. Control is a gene desert region ("Methods"). Dots in bars represent 4 replicates per sample.

their matched wild-type cells ("Methods"). Positive and negative contributions in the regression can be thought of as (positive or negative) "motif activity" that contributes to the differences in gene expression between different lines (Fig. 5c, full table in Supplemental Data 2 and 3). Among the motifs with top motif activity scores in *Mtf2* null cells were motifs that can be bound by GATA1/GATA2, IRX3, and FOXH1, which are known to be crucial for the exit of pluripotency and differentiation toward primitive streak and lateral plate mesoderm progenitors. Also associated with higher expression in *Mtf2* null cells is the presence of motifs for NKX2–5 and HNF4A, which are known regulators of cardiac mesoderm and endoderm. Expression in *Jarid2* null cells was associated with motifs that can be bound by NANOG, members of the PAX family and ectoderm regulators GRHL1/GRHL2 (Fig. 5c).

Next, we wondered if the mRNA expression of these transcription factors correlated with their predicted motif activity (Fig. 5d). *Nanog* was not among the differentially expressed genes. We observed that the mRNA levels of some of these genes, like *Pax8* and *Gli3*, were higher in the *Mtf2* null cells (log2 fold change >1); these genes had similar promoter H3K4me3 levels in wild-type and *Mtf2* null cells, even though H3K27me3 levels at their promoters were reduced in *Mtf2* null cells (Fig. 5d–f). This suggests that, even though these genes were only moderately upregulated upon reduced Polycomb repression, the activity of the motifs bound by them did contribute to stronger activation of genes that were derepressed in the absence of MTF2. Other Polycomb targets such as *Nkx2–5*, *Gata2/3*, and *Irx4* showed a relatively strong derepression with lower H3K27me3 and higher H3K4me3 levels in the undifferentiated *Mtf2* null cells (Fig. 5d–f).

To assess how the loss of *Mtf2* in ESCs would translate into a faster and more pronounced exit of pluripotency, we performed a ChIP of GATA2, which is derepressed in the absence of MTF2 and can bind one of the identified motifs (Fig. 5c, d). Interestingly, we found that GATA2 occupancy was higher at the *Irx3* and *Wnt7b* loci in undifferentiated *Mtf2* null cells (Fig. 5g). Importantly, these PRC2-regulated genes are also higher expressed in the *Mtf2* null cells during the undifferentiated state. This suggests that an MTF2-GATA2 feedforward loop plays a role in the exit of pluripotency by simultaneously reducing repression by Polycomb and stimulating lineage-specific transcription networks and endogenous signaling.

In addition to transcription factors, signaling factors may also influence the exit of pluripotency. We found that many genes involved in signaling pathways such as BMP, TGFβ, and Wnt signaling were upregulated genes in *Mtf2* null cells. Many of these signaling genes also exhibit bivalent promoter marking, and some, for example *Bmp2*, *Wnt7b*, and *Tgfb2*, were more abundantly expressed in *Mtf2* null cells. Upon the loss of *Mtf2*, their promoter H3K27me3 was reduced and H3K4me3 levels were increased in conjunction (Fig. 5d–f). BMP and Wnt signaling is important for specification of lateral mesoderm progenitors[38,39]. Together, our findings revealed that MTF2-mediated recruitment of PRC2 contributes to an epigenomic balance, the loss of which resulted in a stark derepression of lineage transcription factor and signaling genes, and a concomitant increase in activation marks that primes embryonic stem cells toward lineage differentiation.

**Faster progression of *Mtf2* null cells to early germ layer progenitors during directed differentiation**. Finally, we sought to test our findings using an in vitro directed differentiation system. In vivo, the primitive streak gives rise to mesoderm and definitive endoderm via exposure to BMP, TGFβ, and Wnt signals[38,40–42]. We induced the formation of the primitive streak cells from

mESCs over a duration of 24 h and then bifurcated the differentiation based on activation and repression of signals which drive mesoderm and endoderm differentiation, respectively (Fig. 6a).

Cells from different time points (Fig. 6a) were harvested and their bulk-transcriptomes were analyzed. The expression of key regulators of these early differentiation stages like *Nodal*, *Brachyury*, *Bmp2* were indeed progressively elevated, at least in the wild-type cells during these early time points (Fig. 6b, Supplemental Fig. 8).

Next, we compared the differential expression of genes across different time points of *Mtf2* null and *Jarid2* null cells with the wild-type cells. We performed k-means clustering to classify the differentially expressed genes and annotated each of the clusters using Gene Ontology (Fig. 6c, e). Indeed, we observed that the loss of *Mtf2* resulted in upregulation of a group of mesoderm-related genes (Fig. 6c, cluster 6) already at the undifferentiated stage. As differentiation goes on, this group of mesoderm-expressed genes became more prominently expressed, up to 48 h of differentiation. This progressive trend was not observed for the *Jarid2* null cells. Instead, the pattern of expression for *Jarid2* null cells was remarkably similar to wild-type cells.

We observed a similar trend for the endoderm lineage: the activation of endoderm genes was elevated in *Mtf2* null cells from the start (Fig. 6d, f), whereas gene expression in *Jarid2* null cells was similar to that of wild-type cells. It is also noteworthy to point out that non-endoderm gene expression in undifferentiated *Mtf2* null cells (Fig. 6d, for example, clusters 2 and 3), was rapidly curbed during directed differentiation when exogenous factors directing endodermal differentiation were provided. Therefore, while undifferentiated *Mtf2* null cells exhibit multi-lineage derepression of gene expression, these data show that lineage-specific gene expression is successfully established upon lineage commitment in different directions.

Finally, we performed H3K27me3 and H3K4me3 ChIP on the chromatin harvested from both *Mtf2* and *Jarid2* null cells during 48 h of mesoderm differentiation (Fig. 6g, h, Supplemental Fig. 9). We find that, for selected mesendodermal markers such as *Eomes* and *Gata6*, the differences in H3K4me3 and H3K27me3 observed in undifferentiated cells, were exacerbated during mesoderm differentiation. For example, the repressive H3K27me3 levels on *Eomes* were further reduced during differentiation for the *Mtf2* null cells compared to WT and *Jarid2* null cells, accompanied by a dramatic increase in activating H3K4me3 signals (Fig. 6g). This trend is less apparent in *Jarid2* null cells, with a modest increment of H3K4me3 for *Eomes* and a drop for *Gata6* during differentiation (Fig. 6g, h). To connect the phenotypes we observed to a loss in PRC2 function, we treated cells with a chemical inhibitor of EED during differentiation and found that EED inhibition resulted in a decrease in H3K27me3 levels and accompanying increase in H3K4me3 levels on key lineage transcription factors during the differentiation of *Mtf2* null cells (Supplemental Fig. 9b, c). In *Jarid2* null cells, H3K27me3 is similar to the levels in WT cells and is reduced in a similar fashion by EED inhibitor. There is a stronger effect of EED inhibitor on H3K4me3 in these cells in combination with EED inhibitor, in particular on *Eomes* and *Gata6*. These data suggest that the initial epigenetic perturbations caused by the loss of PRC2 subunits in the undifferentiated state directly translate to transcriptional changes during mesoderm differentiation.

## Discussion

Seminal work done on *Mtf2* (PRC2.1) and *Jarid2* (PRC2.2) has highlighted their respective importance in facilitating PRC2 binding to specific genomic loci to compact chromatin[11,15,16,22,43–45]. In the

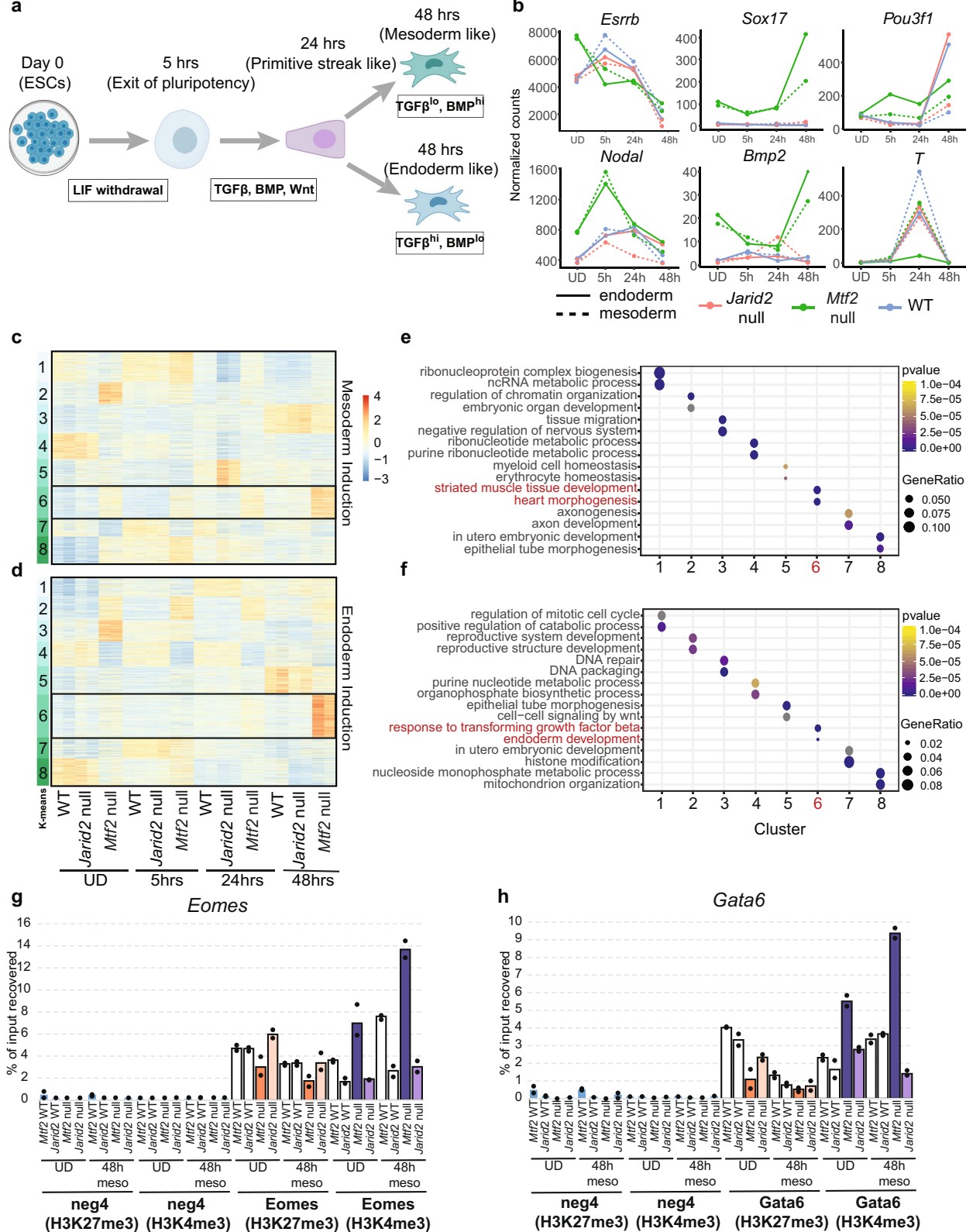

current study, we explored the roles of MTF2 during the exit of pluripotency and discovered that the loss of *Mtf2* alleviated the threshold for differentiation toward lineages in all germ layers, while *Jarid2* null ES cells were partially arrested and slower to differentiate, with a slower downregulation of pluripotency genes. The relatively subtle deregulation in Jarid2 null EBs thus changes their lineage profile, whereas Jarid2 null cells were close to normal in directed

differentiation experiments with exogenous differentiation cues. Importantly, we identified an array of lineage transcription factors and signaling modulators that were upregulated upon the loss of *Mtf2* and that were inherently poised for activation. Upon the loss of *Mtf2*, promoter H3K27me3 levels were markedly reduced, while H3K4me3 was increased. Upon the loss of Jarid2, many genes were also differentially expressed, but a much smaller fraction of the

**Fig. 6 Faster progression of *Mtf2* null cells to early germ layer progenitors during directed differentiation. a** Schematic of directed differentiation for monolayer cells from different genetic backgrounds (WT, *Jarid2* null, and *Mtf2* null). **b** Line plots showing the expression (normalized counts from RNA-seq) of selected temporally regulated genes during early lineage specification processes for different genetic backgrounds and differentiation directions. **c**, **d** Heatmap of differentially expressed genes across all time points and between genetic backgrounds. Data was k-means clustered and the normalized counts were shown. **e**, **f** Dot plots showing the enrichment of biological pathways for each cluster in (**c**, **d**), selected by p-value and gene-ratios of the terms. Top selected pathways were picked for each cluster and shown here. **g**, **h** Barplots of of H3K27me3 and H3K4me3 ChIP for selected targets (*Eomes* and *Gata6*). Each bar represents the average of the percentage of input recovered in the ChIP experiment of replicate experiments. *Mtf2* WT and *Jarid2* WT represent the background-matched wild-type lines of, respectively, *Mtf2* null and *Jarid2* null cells. Each dot represents a qPCR technical replicate for the sample.

upregulated genes in *Jarid2* null cells were EZH2-bound target genes and they were enriched for fewer functional gene ontologies.

Reports on the phenotypic developmental consequences of losing PRC2 subunits during development have been diverse. While the loss of PRC2 core units (SUZ12, EED, and EZH2) have been known to result in gastrulation arrests and impairment of embryo proper differentiation[46], a recent single-cell study done on EED mutant mouse embryos also demonstrated the propensity of these mutants to derive PGC state like cells[47]. Similarly, in our data, we found that our *Eed* mutants overproduce PGC-like cells as well as early differentiation intermediates. The loss of *Mtf2* has been shown to be embryonic lethal by E15.5, with observations of severe anemia and growth retardation[48–50]. By contrast, *Jarid2* null mice exhibited early embryonic lethality (as early as E10.5)[51–53]. To study the mechanisms underlying these disparities between the loss of PRC2 subunits, we adopted the use of embryoid bodies and directed differentiation systems for better control of early differentiation intermediates and signaling paradigms to dissect the roles of PRC2.1 and PRC2.2. Importantly, single-cell RNA sequencing also allows for untangling the heterogeneity of differentiation, uncovering transient developmental events which are difficult to assess in early embryos[54]. For example, we were able to capture aspects of cardiac progenitor development in our dataset (Supplemental Fig. 2a), which could be applicable to the EB field and the formation of gastruloids[55,56].

We found that Wnt signaling was affected upon the loss of *Mtf2*. This is consistent with recent evidence that *Mtf2* null mice demonstrated impairment in definitive erythroid development, and that Wnt was regulated by MTF2 during erythropoiesis[29]. We also found other signaling pathways (e.g., FGF, BMP, and TGFβ) to be deregulated when *Mtf2* is mutated. These pathways were not deregulated in the *Jarid2* null cells, which may potentially explain the delayed and biased EB differentiation we saw in *Jarid2* null cells, whereas it would still allow directed differentiation to proceed when exogenous signaling molecules are supplied.

The PRC2.1 and PRC2.2 complexes functionally interact with each other, as they both depend on positive feedback by binding of core subunit EED to H3K27me3. In the presence of EED inhibitors, PRC2 recruitment becomes much more dependent on JARID2[36]. The JARID2 stoichiometry increases relative to the PRC2 core between pluripotency and differentiated states[57,58]. Conceivably, JARID2 may play a more important role at later stages compared to undifferentiated ES cells.

In addition to signaling factors, we identified key gastrulation transcription factors with bivalent promoters that are poised for transcriptional activation. A reduction of H3K27me3 at their promoters due to the absence of MTF2, resulted in increased levels of H3K4me3. Conversely, since H3K4me3 and H3K27me3 are mutually exclusive on the same histone tail[59], the H3K4me3 permissive mark may also limit the extent to which the promoter can be repressed by PRC2. How this epigenomic balance is regulated is a key question in the field, and our results show that

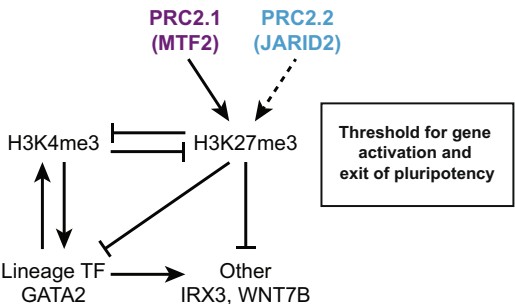

**Fig. 7 Balance of transcriptional activation and exit of pluripotency tilted upon loss of PRC2.1/2.2.** Model of the role of PRC2 in the exit of pluripotency. PRC2 contributes to a repressive threshold for activation of key regulators of differentiation. This affects the exit of pluripotency and the differentiation to lineages of the three germ layers. One example is the upregulation of GATA2 upon MTF2 loss which results in increasing expression of other factors such as WNT7B and IRX3, as part of a feedforward loop that regulates the exit of pluripotency.

MTF2-mediated targeting of PRC2 influences that balance in a way that affects the exit of pluripotency of ES cells. Of note, we also analyzed the shape of the promoters of the previously mentioned signaling and gastrulation factors but did not find a different density of DNA shape-matched GCG sequences, which can be bound by MTF2[15,43]. This suggests that indeed, reduced PRC2 recruitment may be a consequence of stronger activation rather than a reduced capacity for PRC2 recruitment per se. Derepression of poised lineage-specific transcription factors such as GATA2 in turn further contributes to transcriptional activation of key regulators (Fig. 7). This is compatible with recent results, showing that the H3K4 methyltransferase MLL2 protects developmental genes from repression by repelling PRC2[60], highlighting the dynamic epigenomic balance of the classical Trithorax and Polycomb systems.

In summary, our findings revealed the unique differences in differentiation speed and lineage specificity upon the loss of either of the two PRC2 subcomplexes, containing MTF2 (PRC2.1) or JARID2 (PRC2.2). During development, key signaling pathways like BMP, Wnt, FGF, and TGFβ are critically regulated in a time and space-sensitive manner. The activity of these pathways causes lineage specification, but this requires coordinated responses of cells, mediated by a carefully orchestrated balance of activating and repressive cues. Our results in this and a previous study[36] highlight that PRC2.1 and PRC2.2 have distinct contributions in Polycomb repression: PRC2.2 can be recruited by EED binding to H3K27me3 and by PRC1, but on its own affects Polycomb repression in relatively subtle ways, whereas PRC2.1 is directly recruited to select targets that it represses relatively strongly. These differences affect the balance between activation and repression of signaling pathways in highly specific and only partially overlapping ways for PRC2.1 and PRC2.2. This balance is not only important in regulating the threshold for the exit of

pluripotency (Fig. 7), but also during further lineage decisions during development and differentiation.

## Methods

**Mouse embryonic stem cell (mESC) culture**. Wild-type (WT) E14 embryonic stem cells (129/Ola background) were maintained in Dulbecco's modified Eagle medium (DMEM) containing 15% fetal bovine serum, 10 mM Sodium Pyruvate (Gibco), 5 μM beta-mercaptoethanol (BME; Sigma), and Leukemia Inhibitory Factor (LIF: 1000 U/ml; Millipore). The E14 WT[61], Mtf2 null[50], Jarid2 null[7], and Eed null[62] cells used in this paper were maintained in serum + LIF media as described previously. Medium was refreshed once every 2 days. We used background-matched wild-type (WT) cells of Mtf2 null (Pcl1–3 wt ESCs[22]), Jarid2 null (JM8 ESCs[63]), and Eed null (J1 ESCs[62,64]) for bulk RNA sequencing and ChIP analyses.

**mESCs embryoid body (EB) differentiation**. mESCs were dissociated with Accutase[TM] and seeded in Nunclon Sphera 6-well plates (Thermo Fisher Scientific, # 174932) at a cell density of 11,000 cells/well in Serum + LIF medium. After 2 days of aggregation, the embryoid bodies (EB) were let to differentiate by removal of LIF. Differentiation media was refreshed every 2 days by directly pipetting out spent media and adding in new media, with as little disturbance to the EBs as possible. On days of harvest, EBs were pipetted out and spun down at 400 × g for 5 min, followed by dissociation with Accutase[TM] (37 °C, 5 min) and thereafter FACs sorted for viable cells using 7-AAD staining (Thermo Fisher Scientific, #A1310).

**mESCs monolayer differentiation**. mESCs were seeded at a density of 9000 cells/ well of a 12-well cell culture plate. On the next morning, mESCs were washed (once with DMEM) and then differentiated into either anterior primitive streak (APS) by adding 100 ng/mL Activin A (Thermo Fisher Scientific, #PHC9561) + 2 μM CHIR99021 (Peprotech, #2520691) + 20 ng/mL FGF2 (Thermo Fisher Scientific, # 13256029) or mid primitive streak (MPS) (30 mg/mL Activin A + 40 ng/mL BMP4 (Thermo Fisher Scientific, # PHC9533) + 6 μM CHIR99021 + 20 ng/mL FGF2) for 24 h. Subsequently, the cells were further differentiated into definitive endoderm with 100 ng/mL Activin A + 250 nM DM3189 (Peprotech, # 1062443) and lateral mesoderm with 1 μM A-83-01 (Tocris, #2939) + 30 ng/mL BMP4, respectively, for another 48 h. For experiments during directed differentiation toward mesoderm lineage, cells were pre-treated 4 days with a chemical inhibitor of EED (5 μM EED226, Selleckchem, #S8496) before the start of differentiation, and treatment was continued during differentiation, with a change of medium every 2 days.

**RNA extraction and bulk RNA sequencing preparation for monolayer cells**. RNA from mESCs were harvested at several time points (Day 0 undifferentiated, 5 h after induction, 24 h APS/MPS, and 48 h DE/LM). RNA isolation and purification were performed using the Quick–RNA™ MicroPrep kit (Zymo Research), according to the manufacturer's protocol. Integrity of purified RNA was checked on an Agilent Bioanalyzer using the RNA 6000 Pico Kit. Intact RNA was depleted of rRNA and prepared for sequencing with the KAPA RNA HyperPrep Kit with RiboErase (Kapa Biosystems). Libraries were sequenced on the NextSeq 500 (Illumina), generating an average of 12–15 million reads per sample.

**Single-cell RNA library preparation for EBs**. EBs were harvested at different time points (Day 0 undifferentiated, 4, 7, 10 days). The cell suspension was pipetted ~15 times to prevent cell clumping and it was stained with 7-AAD. The live then cells were selected for and FACs-sorted onto 384-well plates containing primers with unique molecular identifiers, according to the SORT-Seq protocol[65]. Plates were spun down (1200 × g, 2 min, 4 °C) and ERCC spike-in mix (1:50,000) was dispensed by a Nanodrop (BioNex Inc) into each well. 150 nl of the Reverse Transcription (RT) mix was similarly dispensed into each well. Thermal cycling conditions were set at 4 °C 5 min; 25 °C 10 min; 42 °C 1 h; 70 °C 10 min. Contents from the plates were pooled together and the cDNA was purified using AmpureXP (New England BioLabs) beads. In vitro transcription (Ambion MEGA-Script) was then carried out overnight at 16 °C, with the lid set at 70 °C. An exonuclease digestion step was performed thereafter for 20 min at 37 °C, followed by fragmentation of the RNA samples. After a beads cleanup, the samples were subjected to library RT and amplification to tag the RNA molecules with specific and unique sample indexes (Illumina), followed by a final beads cleanup (1:0.8, reaction mix: beads) and the sample cDNA libraries were eluted with DNAse free water. Libraries were quantified using by qPCR and sequenced on the NextSeq 500 (Illumina) for 25 million reads per plate.

**Reads filtering, processing, and downstream analyses pipeline**
*Single-cell RNA*. Raw reads were mapped and aligned to the mouse genome GRCm38/mm10 database using the Bowtie2[66] alignment tool. Aligned reads were indexed and the final count table was derived using HTseq[67]. R package 'Scater'[68] was used for sample filtering and quality check, to remove dropouts (cells with <5 reads) and cells with too few recovered genes (<500). The data was then analyzed using R package 'Seurat'[69] (v3) for batch, read counts and gene counts

normalization. In summary, after filtering, we captured the transcriptomes of 1196 WT cells, 1254 Mtf2 null cells, 1196 Jarid2 null cells, and 811 Eed null cells. After filtering, technical confounders such as total number of counts and features were also normalized using the 'LogNormalize' function with a scale factor of 10,000. Next, feature selection was performed using the 'vst' method in Seurat to identify the top 2000 most hypervariable genes (HVGs). Then, the dataset was batch corrected using the linear regression model in built in Seurat to regress out unwanted technical effects of libraries (batches) via scaling. Thereafter, the dataset was subjected to linear dimensional reduction using a principal component analysis of the top 15 principal components (PCs), determined by an elbow plot of the PCs. The cells are then clustered using a shared nearest neighbor (SNN) modularity optimization-based clustering algorithm at a resolution of 2.5 and the 22 different clusters were projected onto a 2-D UMAP for data visualization. Time course trajectory analyses were performed using the package 'Monocle v2'[70]. The clusters information from Seurat were imported into Monocle v2 for analyses and the top 2000 HVG detected in Seurat were used to order the cells during pseudotime analyses. All cells were used in combination for the trajectory analyses and no ground states were set for the pseudotime analysis. Actual time point information and cluster information were overlaid onto the trajectory plots. Cluster identities were defined by Anatomy ontology[71] and correlated with Mouse Cell Atlas[33] (MCA) cluster data. The single-cell data can be downloaded and viewed in a user interface via the following repository: https://github.com/chethloh/ PRC2_singlecelldata.

*Bulk RNA*. Paired-end Illumina 75-bp sequencing files were mapped to the mouse genome GRCm38/mm10 database using the Bowtie2[66] alignment tool. Reads were quantified using Salmon[72] and the count tables were analyzed using DEseq2[73] (version 1.18.1), using Wald statistics (Log2 fold change >1, padj value <0.001) for pairwise comparison and likelihood ratio test statistics (FDR < 0.01) to identify statistically different expression patterns across time points. Gene Ontology enrichment analysis was performed with clusterProfiler[74] (version 3.6.0). Anatomy ontology enrichment was performed MouseMine web interface[71].

*ChIP-seq analyses*. ChIP data for undifferentiated mESCs were generated previously[15]. All fastq files were mapped using bwa (version 0.7.10-r789) and filtered using samtools (version 1.7, flag -F 1024), then normalized for depth of sequencing. Peak-calling was done using MACS2–2.7[75] (q-value < 0.0001). Only peaks that were called in both replicates were used downstream. Heatmaps for ChIP-seq were generated using fluff[76] (v3.0.2) from bam files using read-depth normalization. Reads Per Kilobase of transcript, per Million mapped reads (RPKM) quantification from two independent replicates were performed using scipy (v 1.1.0). GimmeMaelstrom[77] (v 0.14.0) was used for Fig. 5c to identify the transcription factor motifs that are influencing RNA expression dynamics by scanning motifs associated with promoters (+/−0.5 kb from TSS) of a list of differentially regulated genes for both Mtf2 and Jarid2 null cells against their own wild-types at the undifferentiated stage. By default GimmeMaelstrom uses a non-redundant, clustered database of known vertebrate motifs: gimme.vertebrate.v5.0. These motifs come from CIS-BP (http://cisbp.ccbr.utoronto.ca/) and other sources such as JASPAR, IMAGE, HOMER, and Swiss Regulon. DNA shape analysis was performed using the DNAshape package[78]. For Fig. 5b, the peaks that were selected for their respective RPKM value were peaks that are overlapping with peaks in a +/−0.5 kb region around the TSS of the nearest genes that are (1) all PRC2 targets, (2) upregulated PRC2 targets in Mtf2 null cells, and (3) upregulated PRC2 targets in Jarid2 null cells. For Fig. 5g, the ChIP for GATA2 was performed on mouse embryonic stem cells chromatin extract. Chromatin was fixed using 1% formaldehyde for 8 min at r.t. Fixed chromatin was quenched using 1.25 M Glycine solution and sonicated for 8 min of 30 s ON/30 s OFF using the Bioruptor ® Pico sonication device (Diagenode). Sheared DNA was probed with the antibody Anti-GATA-2 Antibody (H-6) (Santa Cruz, #sc-515178, 1:500) overnight. A 10% input control was taken along for each reaction. ChIPped DNA was treated with Protein A/G beads for purification and extensive washes were performed for the DNA, followed by an elution using the MinElute Purification columns (Qiagen). Purified DNA was diluted 4 times with water and probed with PCR primers for Eomes, Pdgfra, Irx3, and Wnt7b genes.

**Antibodies**. ChIP was performed using 3 μl per sample of the following antibodies: MTF2 (Aviva System Biology ARP34292, lot QC49692-42166), H3K27me3 (Millipore 07-449, lot 2717675), EZH2 (Diagenode C15410039, lot 003), H3K4me3 (Ab858, lot GR240214-4), and Anti-GATA-2 Antibody (H-6) (Santa Cruz, #sc-515178, 1:500).

**ChIP-qPCR primer design**. The four primers used for ChIP-qPCR were as follows —Eomes_F: 5′-GATGTCAGCCCGAGTTCTCT-3′, Eomes_R: 5′-ATGGACTTG GATGCTGTGTG-3′, Neg4_F: 5′-AATCCTGAACATGGGAAACCT-3′, Neg4_R: 5′-GGCCTAAGATTCTCTCTTCCATC-3′, Irx3_F: 5′-ACATTTCTACGGGGCC TCAA-3′, Irx3_R: 5′-GACAGGACAGGAGGAGAGTG-3′, Wnt7b_F: 5′-GGTGA CCTGTTCATGTCGAA-3′ and Wnt7b_R: 5′-GTGCTGACCACAGTCCTAAA-3′.

**RT-PCR**. 500 ng of genomic RNA were used for reverse transcription into cDNA using the iScript cDNA synthesis kit (Bio-Rad, #170-8891). cDNA was diluted 10X and 2 μL was added for each qPCR reaction (iQ™ SYBR Green Supermix, Bio-Rad) and subsequently ran on the CFX96 Touch™ Real-Time PCR Detection System.

**Reporting summary**. Further information on research design is available in the Nature Research Reporting Summary linked to this article.

## Data availability

The data that support this study are available from the corresponding author upon reasonable request. The single-cell RNAseq and bulk RNAseq fastq and count matrices have been deposited in the GEO repository under accession code GSE154572. ChIP-seq reads, coverage as genome browser tracks, and peak files have been deposited in the GEO repository under accession code GSE94300. The single-cell data can be downloaded and viewed in a user interface via the following repository: https://github.com/veenstralab/chetloh_ncomm_prc2_singlecell_2021. Source data are provided with this paper.

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

## Acknowledgements

We would like to thank Henk Stunnenberg and Peter Brazda for help with single-cell RNA sequencing at early stages of the project, Eva Janssen-Megens for technical assistance, Guido van Mierlo for scientific input and discussion, and Rob Woestenenk for help with sorting cells.

## Author contributions

C.H.L. and G.J.C.V. designed the experiments and wrote the manuscript. C.H.L. performed all experiments and bioinformatic analyses. M.P. designed experiments in the early phase of the project, provided ChIP data for undifferentiated mESCs, and performed DNA shape prediction analysis. M.R.B. assisted with the bulk RNA-seq analyses. S.v.G. assisted with bulk RNAseq experiments for EB differentiation. Figure 6a scheme was designed by C.H.L.

## Competing interests

The authors declare no competing interests.
