## [Peer Review File · Nature Communications]

REVIEWER COMMENTS

Reviewer #1 (Remarks to the Author):

In this manuscript, entitled "Loss of PRC2 subunits primes lineage choice during exit of pluripotency", the authors dissect the contributions of specific PRC2 accessory components, MTF2 and JARID2, in the regulation of PRC2 activity during embryonic stem cell (ESC) differentiation. Through elegant single-cell RNA-seq experiments they nicely dissect the underlying heterogeneity when ESCs are induced to differentiate to become embryoid bodies. This heterogeneity is a key issue when analysing these types of embryonic stem cell differentiation experiments and the authors nicely address this. The authors also go on to report diverse lineage commitment and specification phenotypes in Mtf2 null vs Jarid2 null ESCs during differentiation. In particular, there is a stronger phenotype in Mtf2 null compared to Jarid2 nulls cells and the authors link this to a more prominent loss of H3K27me3 and concomitant increases in H3K4me3 at the promoters of key developmental regulators upon lineage commitment. While the data presented here is very interesting, there are some outstanding issues with the data and interpretation of results that need clarification.

Major Points:

1. A major issue with this manuscript is the choice of embryonic cell lines, and in particular the fact that the chosen E14 wild-type embryonic stem (ES) cell line does not match the background in which the Mtf2 and Jarid2 genes are deleted. The phenotypic and gene expression variations between the many different wild-type ESC lines means that it will be critical that all experiments are performed in parallel with isogenically matched wild-type ESCs. This is even more critical when analysing the effects of these mutants after induction to differentiate, as slight differences in the chromatin and transcriptional profiles of the different wild-type background cell lines could be exacerbated. While I appreciate it would be a significant undertaking to perform all the scRNA-seq experiments again with proper isogenic matched wild-type cell lines, the authors should at the very least comment on this discrepancy and perform mRNA RT-qPCRs with the proper matched cell lines to confirm the phenotypes observed.
2. The authors nicely include Eed null cells in their analysis as an important control. However, it is slightly surprising that these cells seem to differentiate normally (Fig 1b). This is in stark contrast to previously described differentiation defects observed in Suz12 null ES cells (Pasini et al, MCB 2007). The authors should comment further on this apparent discrepancy and include Suz12 knockout ES line for comparison.
3. Furthermore, the loss of a core PRC2 component and an accessory component during differentiation is a very important comparison which the authors make in Figures 1 and 2. Is the loss of PRC2 targeting as critical as the loss of the whole PRC2 complex? The authors include this critical control in Figures 1 and 2, but leave it out for Figures 3-6. This should be addressed by including Eed ESCs in the analyses, in particular for Figures 3 and 4.
4. Underlying the differentiation defects observed between loss of Mtf2 and Jarid2 is the relative difference in loss of H3K27me3 at PcG target genes. The loss of JARID2 results in a much less pronounced reduction of H3K27me3 than MTF2. Therefore, it is not surprising that loss of MTF2 results in faster differentiation towards all three germ layers. These genes lose more H3K27me3 in MTF2 KO, and therefore will lose more cPRC1 and will be more readily activatable when LIF is removed and induced to differentiate. It is unsurprising that the relative strength in the differentiation phenotype observed correlates with loss of H3K27me3. Having said this, it is interesting that JARID2 loss results specifically in neuro-ectodermal lineage commitment. To further explore this, the authors could perform directed differentiation for this particular lineage in Jarid2, Mtf2 and Eed null ES cells. They could then check JARID2-PRC2.2 and MTF2-PRC2.1 occupancy as well as H3K27me3 enrichment at a selection of these genes. Is this effect specific to loss of PRC2.2?
5. In Figures 4 and 5, the authors link the differentiation effects observed to de-repression of a subset of PcG target genes due to loss of H3K27me3 and concomitant increase in H3K4me3. While this is interesting, it has been shown previously in ES cells. To expand on findings from figures 1-4, the

authors should CHIP H3K27me3 and H3K4me3 during differentiation and match this with their gene expression analysis. This would show direct evidence of loss of a PRC2 accessory component directly affecting gene transcription during differentiation (importantly, this should be done with matched isogenic wild-type cell lines).

6. In figure 5c, the authors suggest via TF binding motif analysis, that certain TFs regulate the exit from pluripotency by activating the genes that lose H3K27me3. Are these TFs PRC2 targets themselves? This result is very intriguing and exciting. Checking the occupancy of these TFs at the genes that lose H3K27me3 would be novel and potentially provide additional insights into the mechanism for the differentiation defects observed.

Minor Points:

1. There are no Figure titles. Please include these.
2. The use and definition of the term "pseudotime" is confusing and difficult to interpret. Please address this adding 1-2 lines in the relevant results section clarifying exactly how this was defined and how it can be interpreted.
3. In figure 4c, there is no indication on the heatmaps as to the size and genomic regions that are in the plot. Are they TSS's/promoter regions? Or is it plotted on all EZH2 peaks centres? How big/broad is the genomic window being plotted?
4. It is important that the authors clarify more clearly that all the experiments done in Figure 5 are in undifferentiated ESCs.

Reviewer #2 (Remarks to the Author):

Loh et al. Loss of PRC2 subunits primes lineage choice during exit of pluripotency.

This study builds off the authors' previous work which addressed the functions of mutually exclusive accessory subunits, Mtf2 and Jarid2, which are found in two different sub-complexes of the polycomb repressor PRC2. By utilizing single-cell and bulk RNA-seq, the authors track the transcriptomes of Mtf2- and Jarid2- null ESCs following embryoid body formation at different time points. The authors perform clustering and ontology grouping to identify the relative germ layer contributions to embryoid bodies for each of their mutants. Additionally, by using a "pseudotime" trajectory analysis, they infer a faster differentiation speed of Mtf2-null cells. To explain these differences in differentiation potential, the authors performed bulk RNA-seq in their mutant cell lines, which showed that KO of Mtf2 led to substantially more transcriptomic changes than KO of Jarid2. Looking at previously generated ChIP-seq data, the authors show that Mtf2-null cells have less H3K27me3 and more H3K4me3 at PRC target genes and, in particular, genes upregulated after Mtf KO, suggesting a role for Mtf2 in keeping a subset of PRC2 targets repressed. Finally, the authors performed endoderm/mesoderm directed differentiation of their Mtf2- and Jarid2- null ESCs and found that Mtf2-null cells progressively upregulated mesoderm related genes in both cases.

Major comments:

While single cell studies of embryoid bodies at different days of differentiation, comparing cell compositions and trajectories of the different mutants with WT cells, is potentially interesting, the significance of this work is diminished by the lack of novel insight regarding the PRC2 subtype-specific role of Mtf2, and the fact that the bottom line of the paper, that loss of PRC2 primes cells for differentiation, is not truly novel, but is rather demonstrated here in another fashion. The authors previously proposed a mechanism for the recruitment and synergy of PRC2.1 and PRC2.2 to chromatin. This study, however, does not provide a deeper understanding of this mechanism nor of the importance of Mtf2 and the specific function of PRC2.1. Importantly, disentangling the consequences of loss of Mtf2 and the defects caused by abnormal PRC2 function in general is of key issue here. The contribution of PRC2 to developmental defects and its epigenetic mechanism is already

well studied. Therefore, in order to make a strong claim as to the role of Mtf2, further comparisons need to be made between Mtf2 mutant cells and PRC2-defective cells (e.g. PRC2 inhibition, knockdown of PRC2 subunits). Further, to what extent do each of the PRC2 sub-complexes themselves contribute to normal development/lineage priming? The lack of transcriptomic changes caused by Jarid2 KO and the redundancy of the two subcomplexes underscores the difficulty of this problem. Also, there are some conclusions (i.e. lack/depletion of a specific germ layer in a specific mutant), which could be tested experimentally by directed differentiation. Finally, some of the conclusions (also regarding MTF2 itself, e.g. Fig.5) are not supported by the data, see below.

Additional comments

The authors have not validated any of their data (scRNAseq nor bulk RNAseq)

Overall, while, as mentioned, the single cell data adds an interesting dimension, its interpretation, especially during the course of differentiation, is far from trivial, especially when trying to compare different cell types with different differentiation propensities. Below are a few examples.

The authors indicate (p.5, top): "Compared to Eed null cells, the transcriptomes of Mtf2 null and Jarid2 null ES cells exhibit smaller differences with wild type ES cells, but still form genotype-specific clusters when not differentiated (clusters 3, 8 and 0 for respectively wild type, Mtf2 null and Jarid2 null cells). In EBs, these undifferentiated cell clusters pass through largely genotype-specific clusters with early differentiation intermediates (respectively 12, 5 and 17), after which they tend to converge into mixed genotype differentiated clusters (Fig. 1c-d, Extended Data Fig. 1b)." Indeed, both clusters 3 & 12 mainly contain WT cells, and both clusters 8 & 5 mainly contain Jarid2 null cells, but cluster 17 does not contain cells from Mtf2 null cells like cluster 0. It appears that the Mtf2 null cells skip the early differentiation intermediates to the next lineage stage.

Also, the authors write (p.5, bottom): "In cluster 5, one of the clusters with early differentiation intermediates enriched for Jarid2 null cells, we observed that the cluster contained cells from both late (day 7 and 10) and earlier time points (Fig. 1d, 2c, Extended Data Fig 2e)." But a close look at the data (e.g. Fig. 2c) reveals that clusters 4 and 21 contain cells from both days 7 and 10, but cluster 5 mainly contains cells from days 4 and 7, and hardly contains cells from day 10. Instead, cluster 17 contains cells from days 7 and 10 for Jarid2. The expression of Otx2 in this cluster is lower.

From Fig. 1c-e it seems that Jarid2 null cells are not represented in day 10 of differentiation. Do the cells fail to differentiate? Do they differentiate slower?

In another instance, the authors write (p.5): "Jarid2 null cells were overrepresented in ectodermal clusters at the expense of mesodermal or endodermal lineages (Fig. 2a). This is mainly on account of Otx2-expressing neuroectodermal cluster 15, in which both Jarid2 null and Mtf2 null cells were overrepresented (Extended Data Fig. 2d). Jarid2 null cells were also overrepresented in clusters 5 and 17, which represent early differentiation intermediates that also express Otx2 (Fig. 1e, g). This combination of Otx2 and pluripotency gene expression, which is also observed in a corresponding cluster of predominantly wild type cells (cluster 12), may correspond to Rosette-stage pluripotency between naive and primed states". When comparing Supplementary Fig. 2d with Figures 1 & 2 it shows that essentially all clusters which contain cells expressing Otx2 have more Mtf2-null cells than Jarid2-null cells, except for cluster 5. But different expression levels in a particular background (e.g. over-expression of Otx2 in Mtf2-null cells) may skew the balance (more cells passing the threshold).

Another confusing issue is Figure 5b. The authors argue that in the absence of H3K27me3, H3K4me3 is elevated, inducing gene expression. But the slight (if any) increase in H3K4me3 in the Mtf2-null cells does not follow the drastic decrease in H3K27me3 in these cells. More importantly, the decrease in H3K27me3 in the up-regulated genes (and concomitant increase in H3K4me3), when compared with 'All PRC2 targets' is observed in BOTH WT and in the Mtf2-null cells, so how can the authors conclude that "Together, these data suggest that the 284 genes are repressed by PRC2 but are poised

for activation, and that PRC2 binding is destabilized by reduced H3K27 methylation, leading to derepression upon the loss of MTF2." (p.10). This effect, it seems, is MTF2-independent.

Overall, among all the details presented in this paper, it is very hard to find the novelty and the take home message.

Minor

The authors indicate 4994 cells after filtering. Please add the number of the cells they started with.

Scalebar is missing in Fig1b.

Supplementary Fig1a – No jarid2 in bulk RNAseq

Metagene profile for H3K4me3/H3K27me3 at PRC2 targets may be a helpful addition to Figure 5.

The Discussion section is a bit unfocused and would greatly benefit from elaboration.

Reviewer #3 (Remarks to the Author):

The manuscript by Loh et al combines single-cell and bulk RNAseq with embryoid body differentiation from ESCs from different genetic backgrounds to highlight different roles of auxiliary PRC2 components (MTF2 and JARID2) in early differentiation: EB from MTF2 null cells are enriched in differentiated cells from all fates, whereas EB from JARID2 null cells are mostly enriched in early differentiating cells or cells in the neuroectodermal fate. The mechanism behind the effect of loss of PRC2 subunits on cell differentiation is studied through a combination of bulk RNA-seq and ChIP-seq in undifferentiated ESCs and in monolayer differentiation. The study, and datasets are novel and will be of interest for the field. However, many aspects of the analysis are not described with enough details to reproduce or review them and a few conclusions require strengthening.

Main points:

A. [Fig1c-e] The current visualization does not allow to distinguish time points and genetic backgrounds (for example, are all Eed null ESCs in the bottom-right cluster?). It will be beneficial to provide UMAPs highlighting individual genotypes over time. Moreover, the authors only describe the central cluster: what are the bottom-left and bottom-right clusters? Are all cells included or were the cells in 'side clusters' excluded in the trajectory analysis?

B. [Fig2] It is currently unclear how meso-, endo- and ectodermal cell types in Fig2a have been defined: the annotations in Fig1e/ text and SupplFig2a do not match (is only cluster 4 defined as Mesodermal or also 13 and 19?), and the description of this section of the analysis is not available in Methods. Authors show number of cells in cluster 4/21/5 to highlight MTF2 null cells' ability to differentiate quickly - however, these numbers are influenced by the total number of cells from that time point/ genetic background in the dataset. What is the relative percentage to the total number of cells from that sample? What is the transcriptional signature of clusters where Mtf2/ Jarid2 null cells are depleted ("Others" in fig2a)?

C. [Fig3] How were genes selected to construct the trajectory? How are the different cell lineages (meso-, endo- and ectodermal) distributed in the trajectory and what are the branching points identified in the trajectory?

D. [Single-cell RNA vs bulk comparison] Authors use both bulk and single-cell RNA-seq to study the effect of the loss of PRC2 subunit on differentiation. To better understand the bulk RNA-seq datasets,

it would be beneficial to know how heterogeneous are the cells in the intermediate time point during differentiation.

E. Authors show the effect of MTF2 loss on differentiation (enhanced and faster differentiation towards all fates) though scRNA-seq and study the effect on endo- and mesodermal fates through monolayer differentiation. It would be interesting to explore the mechanism behind gene upregulation at 48h (cluster 6 in Fig6c and d): are these genes PRC2 targets in ESCs or do the authors think they could be targets of the transcription factors identified in Fig5?

F. It is not clear how the absence of Jarid2 causes cells to differentiate towards early precursors and neuroectodermal fates. Could the marker genes that are upregulated in the single-cell cluster of early differentiating intermediates targeted by PRC2?

G. Please make the single-cell datasets explorable in a Cellxgene or Single-Cell Portal session (or equivalent tools). Computational analyses should be more thoroughly described and code made available on Github.

Other points:

- [Pag 1] Authors should provide the number of cells per genetic background/ time point, and describe the methods utilized for batch and sequencing depth normalization
- [Fig1B] Scale bar is missing
- [Fig1c and e] Clusters in close proximity should have distinguishable colors
- [Pag5, top] "Eed null cells [...] are quite different from wild type ES cells. Within differentiating EBs, however, some of these cells acquire transcriptomes that are similar to cells with wild type or other genotypes, as observed in clusters 19 and 20 for example (Fig. 1c-d, Extended Data Fig. 1b)" Is the knockout fully penetrant or could these cells be wild-type?
- [Pag7] The faster rate of differentiation in Mtf2 null cells is only evident in day 7.
- [Fig4c] Color bar is missing
- [Fig4d and 5b] Provide number of genes in each group in figure legend or plot.
- T/ Brachyury is called differently in Fig 3e and 6b.
- Quantify statements such as "significant number", "large part of the variance" (pag 8) or "relatively strong" (pag 9)
- Clarify in the figure panel that "Up-regulated genes" in Fig 5b are the 284 PRC2 targets and MTF2 upregulated from Fig4b
- [Fig4a, 6c-d] Provide number of genes in heatmap in figure legend or plot.
- [Ext Data Fig1] What was the threshold to call a cluster single or mixed background in (b)? Please also add label to color bar in c-f
- [Ext Data Fig2d-e] Add y axis labels
- [Ext Data Fig3] There is a typo in the title (confirmed)
- [Ext Data Fig3j-l] What is the difference between these panels?
- [Ext Data Fig6a] Second row, last panel – x axis is not horizontal

Revisions and point-by-point response to reviewers

C. Loh et al. (responses indicated with '>>')

Reviewer #1

In this manuscript, entitled “Loss of PRC2 subunits primes lineage choice during exit of pluripotency”, the authors dissect the contributions of specific PRC2 accessory components, MTF2 and JARID2, in the regulation of PRC2 activity during embryonic stem cell (ESC) differentiation. Through elegant single-cell RNA-seq experiments they nicely dissect the underlying heterogeneity when ESCs are induced to differentiate to become embryoid bodies. This heterogeneity is a key issue when analysing these types of embryonic stem cell differentiation experiments and the authors nicely address this. The authors also go on to report diverse lineage commitment and specification phenotypes in *Mtf2* null vs *Jarid2* null ESCs during differentiation. In particular, there is a stronger phenotype in *Mtf2* null compared to *Jarid2* nulls cells and the authors link this to a more prominent loss of H3K27me3 and concomitant increases in H3K4me3 at the promoters of key developmental regulators upon lineage commitment. While the data presented here is very interesting, there are some outstanding issues with the data and interpretation of results that need clarification.

Major Points:

1. A major issue with this manuscript is the choice of embryonic cell lines, and in particular the fact that the chosen E14 wild-type embryonic stem (ES) cell line does not match the background in which the *Mtf2* and *Jarid2* genes are deleted. The phenotypic and gene expression variations between the many different wild-type ESC lines means that it will be critical that all experiments are performed in parallel with isogenically matched wild-type ESCs. This is even more critical when analysing the effects of these mutants after induction to differentiate, as slight differences in the chromatin and transcriptional profiles of the different wild-type background cell lines could be exacerbated. While I appreciate it would be a significant undertaking to perform all the scRNA-seq experiments again with proper isogenic matched wild-type cell lines, the authors should at the very least comment on this discrepancy and perform mRNA RT-qPCRs with the proper matched cell lines to confirm the phenotypes observed.

>>Response: We thank the reviewer for the interest in our work and for thoughtful comments and suggestions. We have repeated the embryoid body (EB) differentiation experiment for all mutants (*Mtf2* null, *Jarid2* null and *Eed* null) with their respective matched wild-type cells. The respective cell lines are described in the updated Methods section, line 399-400. Instead of RT-qPCR, we performed bulk RNA-sequencing on these six lines, at four time points, in duplicate (48 samples). From our bulk-RNA seq analyses with the matched wild-types, we find that *Mtf2* null cells demonstrated up-regulation of mesendodermal genes like *Mixl1* and *Eomes* (new Supplemental Figure 1). This ties in nicely with what we described from the single-cell analyses. Comparatively, *Jarid2* null cells demonstrated a more dampened differentiation towards the two of the germ layers

(comparing with *Mtf2* null), although not completely blocked in any lineages. We observed higher expression of neuronal markers like *Pax6* (new Supplemental Fig 1h). Many neural markers other than *Pax6* (*Sox2*, *Nestin*, *Fgf8*) are not upregulated in *Jarid2* null EB, suggesting that these cells do not have a true neuroectodermal identity. We also find a retention of pluripotent markers after prolonged differentiation (new Supplemental Fig 1k). These findings also strengthen our conclusions based on the single cell data. Furthermore, after repeating the *Eed* null cells differentiation, we observed that differentiation is highly perturbed in these mutants, with impaired differentiation towards mesendodermal lineages and a prominent primordial germ cell (PGC) expression signature (new Supplemental Fig 1c-d). This led us to re-analyze the mutant EBs in the single cell data and compare our initial findings with the validation experiments. Indeed, we find that PGC marker genes were also expressed in *Eed* null cells in the single cell experiment, and most clusters express markers from different lineages. We have rephrased our conclusions concordantly. Last, we find that the matched wild type lines of the *Mtf2* null and *Jarid2* null are relatively similar to each other, whereas the matched wild type line of the *Eed* null cells is different. We analyzed gene expression differences by taking the fold change over matched wild type (new Supplemental Fig. 1b).

>> We agree with the reviewer that the initial differences will have a major impact on differentiation. We therefore used our bulk data to rephrase our conclusions regarding differences in the undifferentiated state relative to their own wild type control, and have focused the subsequent analyses of the manuscript on the comparison between *Mtf2* and *Jarid2*.

>> Furthermore, we have revamped our analysis for Figures 4 and 5 with the new data. In this part of our work we explore the dynamics of H3K27me3 and H3K4me3 levels on promoters of up-regulated genes. We now define these genes using our new data of *Mtf2* and *Jarid2* null cells with their matched WTs (new Fig.4 and Fig.5). The bulk RNAseq validation experiments are shown in the new Supplemental Figure 1 and discussed more in detail in the revised manuscript in lines 89-94 and 130-138, new Fig.4 and Fig.5).

>> Finally, we have also used the matched wild type lines in ChIP experiments (new Fig. 6g-h, new Supplemental Figure 9; cf. point 5 below).

2. The authors nicely include *Eed* null cells in their analysis as an important control. However, it is slightly surprising that these cells seem to differentiate normally (Fig 1b). This in stark contrast to previously described differentiation defects observed in *Suz12* null ES cells (Pasini et al, MCB 2007). The authors should comment further on this apparent discrepancy and include *Suz12* knockout ES line for comparison.

>>Response: We thank the reviewer for this comment. We wish to first clarify that we do not think that the *Eed* null cells are behaving “normally” compared to WT cells. From the single-cell UMAP it is clear that the *Eed* null cells clusters are different compared to the other mutants. For example, cluster 13, 16, which contain later stages, *Eed* cells are clustering separately from the rest of the cells. Additionally, the earlier cells (cluster 6, 9

and 10; bottom right) are also different from the rest of the cells in the dataset. We analyzed the bulk-RNA seq analysis and the results point towards a more PGC-like fate and reduced ability to generate mesendodermal lineages (new Supplemental Fig 1). To reconcile this with our original data, we examined the expression of PGC markers *Dnmt3L*, *Dppa2/3*, and *Dazl*, and found elevated expression in the *Eed* null cells clusters relative to the matched wild type cells (new Supplemental Fig. 3f). This led us to reclassify these *Eed* null cell clusters used for the germ layer enrichment analysis (Figure 2). Clusters 9 and 10, which exhibit an early differentiating / PGC / mixed lineage expression, are removed from the original endoderm classification, leading to a revised panel in Fig 2a to demonstrate more accurately the enrichment of germ layers in *Eed* null cells. Finally, recent data on *Eed* mutant mouse embryos show that the mutants have reduced silencing of PGC related genes such as *Dppa3* and *Esrrb*, and broadly express other PGC genes (Grosswendt, S et al., Epigenetic regulator function through mouse gastrulation. Nature, 584(7819), 102-108), which is concordant with our study. We have made the necessary changes to our text to reflect these findings from the validation experiments (lines 89-94, 130-138).

3. Furthermore, the loss of a core PRC2 component and an accessory component during differentiation is a very important comparison which the authors make in Figures 1 and 2. Is the loss of PRC2 targeting as critical as the loss of the whole PRC2 complex? The authors include this critical control in Figures 1 and 2, but leave it out for Figures 3-6. This should be addressed by including *Eed* ESCs in the analyses, in particular for Figures 3 and 4.

>>Response: We thank the reviewer for raising this point. For Figure 3, where we performed trajectory analyses, our main priority is to highlight the differences in differentiation kinetics between the PRC2 mutant lines during differentiation. We have noted (now added in manuscript) the bias for *Eed* null cells to differentiate towards PGC state, which is present in the undifferentiated state and really different from the behavior of WT, *Mtf2* and *Jarid2* lines. We have therefore focused on the differences in germ layer formation between *Mtf2* and *Jarid2* mutants for the remaining part of the manuscript. However, it is possible to do an integrated analysis of the trajectories with the *Eed* null cells included (new Supplemental Fig. 5). Importantly, including the *Eed* null cells did not change the relative ordering of WT, *Mtf2* and *Jarid2* null cells. Analyzed in this fashion, *Eed* null cells do show fast differentiation kinetics, even higher than *Mtf2* null cells (new Supplemental Fig 5). Unique to the *Eed* null trajectory is the further induction of PGC markers such as *Dppa3*, when compared to the other genotypes (new Supplemental Fig 5). We have included these analyses in the paper.

4. Underlying the differentiation defects observed between loss of *Mtf2* and *Jarid2* is the relative difference in loss of H3K27me3 at PcG target genes. The loss of JARID2 results in a much less pronounced reduction of H3K27me3 than MTF2. Therefore, it is not surprising that loss of MTF2 results in faster differentiation towards all three germ layers. These genes lose more H3K27me3 in MTF2 KO, and therefore will lose more cPRC1 and will be more readily activatable when LIF is removed and induced to differentiate. It is unsurprising that the relative strength in the differentiation phenotype observed correlates with loss of H3K27me3. Having

said this, it is interesting that JARID2 loss results specifically in neuro-ectodermal lineage commitment. To further explore this, the authors could perform directed differentiation for this particular lineage in *Jarid2*, *Mtf2* and *Eed* null ES cells. They could then check JARID2-PRC2.2 and MTF2-PRC2.1 occupancy as well as H3K27me3 enrichment at a selection of these genes. Is this effect specific to loss of PRC2.2?

>>Response: While we find that *Jarid2* null EBs are enriched for neuroectodermal cells, we would like to stress that *Jarid2* null EBs are not exclusively making this cell type. We should also mention that all three PRC2 mutants show enrichment for neuroectodermal cells. The difference between *Mtf2* null and *Jarid2* null cells is that *Mtf2* EBs are also enriched for mesoderm and endoderm, whereas *Jarid2* is not. PRC2 mutant cells overexpress a number of neuroectodermal regulators (for example Pax6) even in their undifferentiated state. *Mtf2* null cells additionally overexpress mesendodermal regulators. We have performed 48hrs mesodermal differentiation and analyzed the levels of H3K27me3 and H3K4me3 at a neuroectodermal marker, *Ascl1*. We saw that the levels of repressive mark H3K27me3 were lost in *Jarid2* null and *Mtf2* null cells compared to WT, and that this effect increased during differentiation (new Supplemental Fig 9a).

5. In Figures 4 and 5, the authors link the differentiation effects observed to de-repression of a subset of PcG target genes due to loss of H2K27me3 and concomitant increase in H3K4me3. While this is interesting, it has been shown previously in ES cells. To expand on findings from figures 1-4, the authors should CHIP H3K27me3 and H3K4me3 during differentiation and match this with their gene expression analysis. This would show direct evidence of loss of a PRC2 accessory component directly affecting gene transcription during differentiation (importantly, this should be done with matched isogenic wild-type cell lines).

>>Response: Because of our focus on the exit of pluripotency, we examined the inherent (undifferentiated) de-repression differences between the PRC2 subunit which could lead to the phenotypes observed. To illustrate the outcome of losing a PRC2 subunit during differentiation, we repeated directed differentiation of both the *Mtf2* and *Jarid2* null cells towards a mesodermal fate for 48 hours using the protocol described in the methods section. We included the necessary isogenic WT cell lines as controls and added in a treatment control of a chemical inhibitor of the EED protein that causes PRC2 to lose its activity. This treatment with EED226 is described in the methods section (line 413-416). We performed CHIP-qPCR for H3K27me3 and H3K4me3 on these cells and tested the relative abundance of these marks on selected key PRC2 targets from different germ layers. The results can be found in the new Fig 6g-h and Extended Data Fig. 9. In summary, we found that key transcription factors such as *Gata6* and *Eomes* have reduced H3K27me3 and concomitantly increased H3K4me3 upon the loss of *Mtf2* (Fig 6g-h) and that these differences are exacerbated during differentiation / induction of signaling towards mesodermal progenitors. Interestingly, we found that adding EED226 enhanced the switch from repression to activation during differentiation, further corroborating that the phenotype we observe is a consequence of losing MTF2 and PRC2 functions (line 327-338).

6. In figure 5c, the authors suggest via TF binding motif analysis, that certain TFs regulate the exit from pluripotency by activating the genes that lose H3K27me3. Are these TFs PRC2 targets themselves? This result is very intriguing and exciting. Checking the occupancy of these TFs at the genes that lose H3K27me3 would be novel and potentially provide additional insights into the mechanism for the differentiation defects observed.

>>Response: We thank the reviewer for raising this interesting point. The TF binding motif analyses indeed demonstrate several key TFs that are influencing the expression of the genes that are deregulated upon *Mtf2* loss. Some of these (*Gata2*, *Pax8*, *Nkx2-5*) are PRC2 targets themselves (as evidenced by H3K27me3, Fig. 5e), whereas others (*Gata1*) are not and may contribute indirectly by activating derepressed PRC2 target genes. In our revision experiments, we also performed CHIP qPCR analyses for *Gata6*, one of these TFs that are upregulated in *Mtf2* null cells. We saw that the *Gata6* loses its repressive regulation (H3K27me3) and gains H3K4me3 levels in both undifferentiated cells and 48hrs of differentiation (new Fig 6h). *Gata6* mRNA levels are also higher in *Mtf2* null cells (Fig. 5d). As *Gata6* is also a key mesendodermal TF, this highlights that some of the TFs are derepressed PRC2 target themselves.

Minor Points:

1. There are no Figure titles. Please include these.

>>Response: The figure titles can be found in the individual figure files. The panels in the main manuscript text have been altered to include also figure titles.

2. The use and definition of the term “pseudotime” is confusing and difficult to interpret. Please address this adding 1-2 lines in the relevant results section clarifying exactly how this was defined and how it can be interpreted.

>>Response: The pseudotime analyses are further described in the results section in line 157-158. Briefly, cells are ordered based on gene expression differences. In dynamic processes such as differentiation, these changes are progressive in nature and happen over time. Because the real time scale of the expression changes is unknown (based on expression data alone), this is conceptualized as pseudotime. What we have done in our paper is to compare pseudotime (as a proxy for distance from the undifferentiated state) to real time (as we sampled at different days). The benefit of the single cell trajectory analysis is that, while differentiation progress is heterogeneous, even among cells with the same genotype, the ordering of cells based on their gene expression profiles allows to resolve their progress on the single cell level.

3. In figure 4c, there is no indication on the heatmaps as to the size and genomic regions that are in the plot. Are they TSS's/promoter regions? Or is it plotted on all EZH2 peaks centres? How big/broad is the genomic window being plotted?

>>Response: We have included in the legend to indicate genomic window (+/- 5kb around TSS of the genes)

4. It is important that the authors clarify more clearly that all the experiments done in Figure 5 are in undifferentiated ESCs.

>>Response: We have included figure titles for Fig. 5b to clarify that the genes involved were taken from the undifferentiated ESCs as previously described in Fig. 4.

Reviewer #2

Loh et al. Loss of PRC2 subunits primes lineage choice during exit of pluripotency.

This study builds off the authors' previous work which addressed the functions of mutually exclusive accessory subunits, Mtf2 and Jarid2, which are found in two different sub-complexes of the polycomb repressor PRC2. By utilizing single-cell and bulk RNA-seq, the authors track the transcriptomes of Mtf2- and Jarid2- null ESCs following embryoid body formation at different time points. The authors perform clustering and ontology grouping to identify the relative germ layer contributions to embryoid bodies for each of their mutants. Additionally, by using a "pseudotime" trajectory analysis, they infer a faster differentiation speed of Mtf2-null cells. To explain these differences in differentiation potential, the authors performed bulk RNA-seq in their mutant cell lines, which showed that KO of Mtf2 led to substantially more transcriptomic changes than KO of Jarid2. Looking at previously generated ChIP-seq data, the authors show that Mtf2-null cells have less H3K27me3 and more H3K4me3 at PRC target genes and, in particular, genes upregulated after Mtf2 KO, suggesting a role for Mtf2 in keeping a subset of PRC2 targets repressed. Finally, the authors performed endoderm/mesoderm directed differentiation of their Mtf2- and Jarid2- null ESCs and found that Mtf2-null cells progressively upregulated mesoderm related genes in both cases.

Major comments:

While single cell studies of embryoid bodies at different days of differentiation, comparing cell compositions and trajectories of the different mutants with WT cells, is potentially interesting, the significance of this work is diminished by the lack of novel insight regarding the PRC2 subtype-specific role of Mtf2, and the fact that the bottom line of the paper, that loss of PRC2 primes cells for differentiation, is not truly novel, but is rather demonstrated here in another fashion. The authors previously proposed a mechanism for the recruitment and synergy of PRC2.1 and PRC2.2 to chromatin. This study, however, does not provide a deeper understanding of this mechanism nor of the importance of Mtf2 and the specific function of PRC2.1. Importantly, disentangling the consequences of loss of Mtf2 and the defects caused by abnormal PRC2 function in general is of key issue here. The contribution of PRC2 to developmental defects and its epigenetic mechanism is already well studied.

Therefore, in order to make a strong claim as to the role of Mtf2, further comparisons need to be made between Mtf2 mutant cells and PRC2-defective cells (e.g. PRC2 inhibition, knockdown of PRC2 subunits). Further, to what extent do each of the PRC2 sub-complexes themselves contribute to normal development/lineage priming? The lack of transcriptomic changes caused by *Jarid2* KO and the redundancy of the two subcomplexes underscores the difficulty of this problem. Also, there are some conclusions (i.e. lack/depletion of a specific germ layer in a specific mutant), which could be tested experimentally by directed differentiation. Finally, some of the conclusions (also regarding MTF2 itself, e.g. Fig.5) are not supported by the data, see below.

>>Response: We thank the reviewer for the interest and constructive criticism. At several points we do now present evidence on how the contributions of MTF2 and JARID2 relate to each other and to the function of PRC2, both in terms of molecular effects and differentiation potential. Overall the impact of derepression of gene expression is *Eed* > *Mtf2* > *Jarid2*. Derepression by loss of *Jarid2* tends to be less severe in terms of fold change, but is also more restricted in terms of functional categories. Also, the overlap between genes derepressed by *Mtf2* and *Jarid2* is modest, indicative of functional complementarity. Notably, key mesendodermal regulators are not or substantially less derepressed in *Jarid2* null cells. The response to other comments is found below.

Additional comments

The authors have not validated any of their data (scRNAseq nor bulk RNAseq)

>>Response: In the context of this comment we understand validation to refer to confirmation with another technology or technique. All techniques have pros and cons. Single cell omics resolves cellular heterogeneity very powerfully, but has limitations in terms of sensitivity. Bulk RNA-sequencing is quantitative and sensitive, but is lacking single cell resolution. RT-qPCR is extremely sensitive with high accuracy, but features low precision, very low throughput (with a potential for bias), and no single cell resolution.

>>To further corroborate the observations we made from the single cell RNA-seq, we have repeated the experiments of EB differentiation using the mutants (*Mtf2*, *Jarid2* and *Eed* null cells) matched with their respective WT cells. We performed bulk RNAseq analyses for this experiment covering up to 7 days of differentiation and the data is shown in the new Supplemental Fig.1. Essentially, we confirm that the loss *Mtf2* results in an all-round increase in differentiation efficiency towards all three germ layers, particularly for mesendodermal lineages (new Supplemental Fig.1). *Jarid2* null cells, on the other hand, demonstrated a much lower efficiency differentiating towards mesoderm and endoderm. They appear to be more stagnated during differentiation, as shown by the prolonged expression of undifferentiated markers like *Nanog* (new Supplemental Fig 1k). Similarly, *Eed* null cells appear to be aberrant in their differentiation, particularly towards the primitive streak/mesendodermal lineages. We do note the upregulation of genes related to PGC differentiation, such as *Dppa2*, *Nanog*, *Dnmt3l* (Supplemental Figs. 1j, 3f). This is in line with recent data on *Eed* mutant mouse embryos, showing that mutants have reduced

silencing of PGC related genes such as *Dppa3* and *Esrrb*, and broadly express other PGC genes (Grosswendt, S et al., Epigenetic regulator function through mouse gastrulation. Nature, 584(7819), 102-108). Overall, this new data allowed us to support our initial findings about the differentiation efficiencies between the PRC2.1/2 mutants and provided new insights for us to conclude some phenotypes we saw in the *Eed* null mutants. Furthermore, with the matched WT bulk RNA-seq experiments, we found a set of up-regulated genes for *Jarid2* null cells which we have since included in the new Figure 4 to characterize the phenotypes of the *Jarid2* null cells more accurately. We found that loss of *Jarid2* leads to a more selective group of genes being deregulated, whereas *Mtf2* loss leads to a broader deregulation of PRC2 targets. We also repeated our motif activity analysis to include the genes that are deregulated upon *Jarid2* loss. We have made these changes in the revised manuscript in the following sections: 89-94, 130-138, new Fig. 4 and Fig. 5.

>>Secondly, to further explore and untangle the effects of PRC2 and their subunits during differentiation, we additionally performed directed differentiation of *Mtf2* and *Jarid2* null mutants with their isogenically matched WT cells, towards mesoderm progenitors for two days. We followed this up with a ChIP-qPCR analysis of the repressive H3K27me3 and activating H3K4me3 levels of key transcription factors (TFs) involved during early differentiation, such as *Gata6* and *Eomes*. We found that the initial observations we have at undifferentiated state still stand, that these key TFs already have lowered levels of H3K27me3 in *Mtf2* null cells, compared to *Jarid2* null. During differentiation, this effect is further exacerbated for *Mtf2* null cells, suggesting that the initial de-repression of the TFs, coupled with upregulation of signaling factors further enhanced differentiation towards mesoderm lineage for *Mtf2* null embryoid bodies. To investigate the mechanism driving this differentiation propensity, we also performed the directed differentiation with the addition of a chemical inhibitor of EED (EED226). This inhibition of PRC2 activity resulted in an enhancement of the effects we saw from the undifferentiated state and during differentiation, suggesting that the differentiation phenotypes we observed in *Mtf2* null cells were linked to the loss of PRC2 core activity. The new ChIP data can be found in the new Fig. 6 g-h and Supplemental Fig. 9.

>>These additional data validate and further corroborate many of our conclusions. Moreover, our new analyses shed more light on the phenotypes observed in different mutants. The *Eed* null cells show many strongly upregulated genes, which tend to be substantially less upregulated in *Jarid2* null cells, whereas *Mtf2* null cells show strong deregulation of a partially overlapping set of genes (new Supplemental Figure 1a).

Overall, while, as mentioned, the single cell data adds an interesting dimension, its interpretation, especially during the course of differentiation, is far from trivial, especially when trying to compare different cell types with different differentiation propensities. Below are a few examples.

The authors indicate (p.5, top): "Compared to Eed null cells, the transcriptomes of *Mtf2* null and *Jarid2* null ES cells exhibit smaller differences with wild type ES cells, but still form genotype-specific clusters when not differentiated (clusters 3, 8 and 0 for respectively wild type, *Mtf2* null and *Jarid2* null cells). In EBs, these undifferentiated cell clusters pass through largely genotype-specific clusters with early differentiation intermediates (respectively 12, 5 and 17), after which they tend to converge into mixed genotype differentiated clusters (Fig. 1c-d, Extended Data Fig. 1b)."

Indeed, both clusters 3 & 12 mainly contain WT cells, and both clusters 8 & 5 mainly contain *Jarid2* null cells, but cluster 17 does not contain cells from *Mtf2* null cells like cluster 0.

It appears that the *Mtf2* null cells skip the early differentiation intermediates to the next lineage stage.

>>Response: We are happy to clarify this point. We note that cluster 17 contains mostly *Jarid2* null cells, but it is not completely without *Mtf2* null cells (100 *Jarid2* null cells, 25 *Mtf2* null cells). Furthermore, in another differentiating intermediate cluster (cluster 5), we also captured 63 *Mtf2* null cells compared to 251 *Jarid2* null cells. Overall, we did capture *Mtf2* null cells in all the clusters that we classify as early differentiating intermediates (clusters 5, 7, 12 and 17, Supplemental Table 1). The point is that the numbers of *Mtf2* null cells are much lower compared to other genotypes, especially at later time points (Fig 2c right panel, Suppl. Fig. 3e). So they are not skipping the stage of differentiation intermediates but are differentiating faster (Fig. 3).

Also, the authors write (p.5, bottom): "In cluster 5, one of the clusters with early differentiation intermediates enriched for *Jarid2* null cells, we observed that the cluster contained cells from both late (day 7 and 10) and earlier time points (Fig. 1d, 2c, Extended Data Fig 2e)." But a close look at the data (e.g. Fig. 2c) reveals that clusters 4 and 21 contain cells from both days 7 and 10, but cluster 5 mainly contains cells from days 4 and 7, and hardly contains cells from day 10. Instead, cluster 17 contains cells from days 7 and 10 for *Jarid2*. The expression of *Otx2* in this cluster is lower.

>>Response: We wish to clarify the message for Fig 2c. Indeed, we agree that cluster 5 contains more cells from early than from late time points. The point that we make, however, is that some *Jarid2* cells are lingering in this cluster (Fig 2c) as well as cluster 17 (*Jarid2*-dominated, Suppl. Fig. 3e), and to a lesser extent clusters 12 and 7 (WT-dominated, Suppl. Fig. 3e). So, while quite some *Jarid2* null cells and some WT cells from late time points (day 10 and 7) cluster with cells from earlier time point cells, this does hardly happen with *Mtf2* null cells. That day 10 cells are lingering in the early progenitor state which is also apparent from Fig. 1d.

>>In Fig. 2c and Suppl. Fig. 3e it can be seen that *Jarid2* null EBs contribute a large share of these day 10 early progenitor cells. For a comprehensive picture of differentiation intermediates, one needs to consider all clusters containing these intermediates: clusters 5, 17, 12 and 7. WT, *Mtf2* null and *Jarid2* null cells contribute respectively 487, 112 and 359 cells to these clusters (Supplemental Table 1). Whereas, cluster 12 (and to a lesser extent cluster 7) contains WT day 10 cells, cluster 17 (and to a lesser extent cluster 5) contains many *Jarid2* null day 10 cells (Fig. 2c, Extended Data Fig. 3e). None of the clusters contains a significant number of *Mtf2* null day 10 cells. Our conclusion therefore is that we observe a relatively large contribution of *Jarid2* day 10 cells to clusters of early differentiation intermediates, and only a small contribution of *Mtf2* null day 10 cells, pointing towards a less efficient differentiation for *Jarid2* compared to *Mtf2* null cells.

From Fig. 1c-e it seems that *Jarid2* null cells are not represented in day 10 of differentiation. Do the cells fail to differentiate? Do they differentiate slower?

>>Response: *Jarid2* null cells are represented in our day 10 data, contributing to cluster 17 for example. The total numbers of cells were comparable as well including at the later time points (Supplemental Table 1; Suppl. Fig. 5). We have performed pseudotime analyses, which capture the state of differentiation of every single cell based on their transcriptomes (Fig 3, Suppl. Fig. 5). The results show that *Jarid2* null cells are differentiating at a slower speed compared with *Mtf2* null cells, particularly during the mid-differentiation stages between day 4 to day 7. They still manage to differentiate into cells from different germ layers, albeit at lower efficiencies compared to *Mtf2* null cells. This is also corroborated by our additional validation experiments done with bulk RNAseq on the different mutants with genetically matched WTs, showing a reduced down-regulation of pluripotent markers in *Jarid2* null cells compared to matched wild types, even at later time points of differentiation (Supplemental Fig. 1k).

In another instance, the authors write (p.5): “*Jarid2* null cells were overrepresented in ectodermal clusters at the expense of mesodermal or endodermal lineages (Fig. 2a). This is mainly on account of *Otx2*-expressing neuroectodermal cluster 15, in which both *Jarid2* null and *Mtf2* null cells were overrepresented (Extended Data Fig. 2d). *Jarid2* null cells were also overrepresented in clusters 5 and 17, which represent early differentiation intermediates that also express *Otx2* (Fig. 1e, g). This combination of *Otx2* and pluripotency gene expression, which is also observed in a corresponding cluster of predominantly wild type cells (cluster 12), may correspond to Rosette-stage pluripotency between naïve and primed states”. When comparing Supplementary Fig. 2d with Figures 1 & 2 it shows that essentially all clusters which contain cells expressing *Otx2* have more *Mtf2*-null cells than *Jarid2*-null cells, except for cluster 5. But different expression levels in a particular background (e.g. over-expression of *Otx2* in *Mtf2*-null cells) may skew the balance (more cells passing the threshold).

>>Response: In the germ layer analysis, we included clusters 1, 11, 15 and 16 in the (neuro)ectodermal lineages for the enrichment comparison between mutants. Clusters 11 and 15, which do express *Otx2*, show a higher number of *Mtf2* null cells than *Jarid2* null cells. In cluster 1 it is about the same (91 and 101 cells in resp. *Mtf2* and *Jarid2* null),

whereas cluster 16 only contains a few *Mtf2* and *Jarid2* null cells (Supplemental Table 1). These are not the only clusters with *Otx2* expression, however. The early differentiation intermediates (clusters 5, 17, 12, 7) exhibit *Otx2* expression and show more *Jarid2* null cells than *Mtf2* null cells (Supplemental Table 1; see also response to points 3-5). This supports our interpretation: Both *Jarid2* and *Mtf2* null EBs produce more (neuro)ectodermal cells than WT EBs, *Jarid2* null EBs produce more early intermediates, including cells from later time points, whereas *Mtf2* null EBs produce more mesendodermal cells.

>>This analysis is done at the level of clusters of single cells. These clusters are groups of cells with similar expression. We simply determine the distribution of genotypes and time points over the clusters. So we do not set a threshold for *Otx2* expression, we simply note that some clusters do express *Otx2*, whereas in other clusters this is not detected. At the level of individual single cells it is possible that *Otx2* expression remains undetected (due to drop-out, a known issue with single cell sequencing), but at the cluster level this is less likely because of the number of cells present in the clusters.

Another confusing issue is Figure 5b. The authors argue that in the absence of H3K27me3, H3K4me3 is elevated, inducing gene expression. But the slight (if any) increase in H3K4me3 in the *Mtf2*-null cells does not follow the drastic decrease in H3K27me3 in these cells. More importantly, the decrease in H3K27me3 in the up-regulated genes (and concomitant increase in H3K4me3), when compared with 'All PRC2 targets' is observed in BOTH WT and in the *Mtf2*-null cells, so how can the authors conclude that "Together, these data suggest that the 284 genes are repressed by PRC2 but are poised for activation, and that PRC2 binding is destabilized by reduced H3K27 methylation, leading to derepression upon the loss of MTF2." (p.10). This effect, it seems, is MTF2-independent.

>>Response: The new analysis in Figure 5 compares all PRC2 targets to the subset of PRC2 targets that gets transcriptionally derepressed by the absence of MTF2 and JARID2. In order to link the differentiation phenotypes to PRC2 function, we examined the derepressed genes in more detail and compared them to the larger set of all EZH2-bound genes. This is to assess if there is an aspect of PRC2 activity that is different for the subset of genes that is derepressed.

>>Our take-home conclusion is that the molecular aspects of PRC2 recruitment and activity are very similar for these subsets of genes (All EZH-bound genes versus derepressed EZH2-bound genes). This take-home conclusion has remained the same, however, with one difference: Because we now define derepressed genes with new data, in which we compare the mutant lines to their matched background control, the set of genes is slightly different (242 genes, previously 284). Some of the subtle differences in H3K27me3 and H3K4me3, that the reviewer noticed have become smaller. What is consistent, very robust and statistically significant, is the increase in H3K4me3 upon reduction of H3K27me3. This is observed globally (Fig. 5b), but genes differ in the extent to which they show an increase in H3K4me3. We therefore further explored the role of (poised) activation in derepression. The results suggest that these activating influences, which indeed are MTF2-independent, are normally repressed by MTF2-PRC2 (therefore 'poised'). In the absence of MTF2, the

activating influences are no longer kept in check by PRC2, and the genes become derepressed and transcriptionally active.

>>To further corroborate this conclusion, we have repeated the directed differentiation of the mutant and WT lines to the mesoderm lineage and analyzed the levels of H3K27me3 and H3K4me3 of selected markers during differentiation. We saw that the decrease in H3K27me3 levels and increase in H3K4me3 levels were exacerbated in *Mtf2* null cells compared to WT during differentiation for key mesendodermal markers like *Gata6* and *Eomes*.

Overall, among all the details presented in this paper, it is very hard to find the novelty and the take home message.

>>Response: We have improved the paper in highlighting the most important points in the paper. Briefly: The novelty of the paper is that we dissect the biological roles of PRC2 mutants and link differences in differentiation bias and kinetics to specific derepressed subsets of PRC2 targets and the extent of derepression. Overall JARID2-dependent PRC2 activity is more restricted compared to MTF2 in both the subset of genes that are derepressed and the extent of their extent of derepression. As a consequence of this, the biological effects of their loss are different even though the two PRC2 complexes work together. Complex biological differences between PRC2 mutants arise from differential loss of PRC2-mediated repression that affects the three germ layers in different ways.

Minor:

The authors indicate 4994 cells after filtering. Please add the number of the cells they started with.

>>Response: The exact number of cells before filtering was 5454. We now state the global number (“over 5400”) in line 81 of revised manuscript.

Scalebar is missing in Fig1b.

>>Response: Added 200 um scale bar to revised figure.

Supplementary Fig1a – No jarid2 in bulk RNAseq

>>Response: We have replaced the bulk RNA seq validation with a new experiment for the revision. The new experiment contains all mutants with their respectively matched WT cells (new Supplemental Figs. 1, 6; Figs. 4, 5).

Metagene profile for H3K4me3/H3K27me3 at PRC2 targets may be a helpful addition to Figure 5.

>>Response: A box plot visualizes the same information (peak intensities) as meta-gene profiles over a window around the peak. This quantitative information is provided in Fig. 5b and Supplemental Fig. 7a.

The Discussion section is a bit unfocused and would greatly benefit from elaboration.

>>Response: We have made several additions to the discussion, to elaborate on some of our findings of the mutants with recent data on mouse embryos.

Reviewer #3

The manuscript by Loh et al combines single-cell and bulk RNAseq with embryoid body differentiation from ESCs from different genetic backgrounds to highlight different roles of auxiliary PRC2 components (MTF2 and JARID2) in early differentiation: EB from MTF2 null cells are enriched in differentiated cells from all fates, whereas EB from JARID2 null cells are mostly enriched in early differentiating cells or cells in the neuroectodermal fate. The mechanism behind the effect of loss of PRC2 subunits on cell differentiation is studied through a combination of bulk RNA-seq and ChIP-seq in undifferentiated ESCs and in monolayer differentiation. The study, and datasets are novel and will be of interest for the field. However, many aspects of the analysis are not described with enough details to reproduce or review them and a few conclusions require strengthening.

>>Response: Thank you for your interest in our work and your comments to further strengthen the conclusions.

Main points:

A. [Fig1c-e] The current visualization does not allow to distinguish time points and genetic backgrounds (for example, are all *Eed* null ESCs in the bottom-right cluster?). It will be beneficial to provide UMAPs highlighting individual genotypes over time. Moreover, the authors only describe the central cluster: what are the bottom-left and bottom-right clusters? Are all cells included or were the cells in 'side clusters' excluded in the trajectory analysis?

>>Response: We are happy to provide additional UMAP visualizations for individual genotypes over time in Supplemental Figure 2. To further elaborate on this, Supplemental Fig. 2b shows indeed that the bottom right clusters (6, 9 and 10) belong exclusively to one genotype (*Eed* null). They represent cells from the undifferentiated state with expression of *Esrrb* and cells from later stages resembling PGC-like cells (Supplemental Fig 1e and 3f; new). The bottom left clusters (18 and 19) are annotated as hematopoietic progenitors and endothelial cells derived from mesoderm respectively (Supplemental Fig 3a and b). We have now included plots highlighting cells from clusters 18 and 19 in Supplemental Fig 4 for further clarification.

>>All cells (4994) were included for the trajectory analyses of Supplemental Figure 5 (with *Eed* null cells included). All cells were included in the trajectory analysis of Fig. 3, except for the *Eed* null cells.

B. [Fig2] It is currently unclear how meso-, endo- and ectodermal cell types in Fig2a have been defined: the annotations in Fig1e/ text and SupplFig2a do not match (is only cluster 4 defined as Mesodermal or also 13 and 19?), and the description of this section of the analysis is not

available in Methods. Authors show number of cells in cluster 4/21/5 to highlight MTF2 null cells' ability to differentiate quickly - however, these numbers are influenced by the total number of cells from that time point/ genetic background in the dataset. What is the relative percentage to the total number of cells from that sample? What is the transcriptional signature of clusters where *Mtf2*/ *Jarid2* null cells are depleted (“Others” in fig2a)?

>>Response: We are happy to clarify Fig2a and the analyses that follow. The germ layer annotations, as shown in Supplemental Fig 3c, were derived by selecting clusters which contain cells that were enriched for broad endoderm, mesoderm and ectodermal subtypes, based on the anatomy ontology performed. Therefore, clusters 4, 13 and 19 are all broadly categorized as mesodermal subtypes. In Fig2a, the “others” category refers to cells that are either pluripotent or early *Otx2* expressing precursors.

>>Regarding the influence of the total number of cells per background or time point on our calculations, we would like to point out that the calculation of fold enrichment (Fig. 2a) is normalized for cell numbers (cf. Supplemental Table1). Second, the cell numbers are quite similar (~1200 cells per genotype) except for *Eed* null cells which was a bit lower (811 cells; Supplemental Table 1). For Fig. 2c we plot absolute numbers. Indeed, these numbers might be influenced by the total number of cells from the respective clusters shown. This panel merely serves to highlight that there are more *Mtf2* null cells which are found in these mesodermal clusters compared to *Jarid2* null cells and that this is suggestive of a more efficient differentiation of *Mtf2* null cells towards those lineages. The kinetics of *Mtf2* differentiation was analyzed further using trajectory analyses in Fig. 3. We have rephrased the text to elaborate these points and have included the definition criteria of our clusters in Fig. 2a from line 119 – 122 in the revised manuscript.

C. [Fig3] How were genes selected to construct the trajectory? How are the different cell lineages (meso-, endo- and ectodermal) distributed in the trajectory and what are the branching points identified in the trajectory?

>>Response: The genes that were used for the trajectory were derived from hyper-variable gene testing function in the Seurat package (which is also used for clustering of cells). This is done using the variance relative to the mean expression. Hypervariable means that they are more variable between cells than expected based on their mean expression; this strongly selects for markers of cell types or cell states. Supplemental Fig. 4e-j visualizes the relationships between germ layers and genotypes on the one hand, and the trajectories on the other. We have added some figure annotations in the revised Supplemental Fig. 4. Trajectory analysis, which we performed using Monocle2, orders cells in pseudo-time, based on the global similarity of single cell transcriptomes. The trajectory of cells over pseudotime is a measure of (dis)similarity. Importantly, pseudotime (which is based purely on transcriptome data) broadly recapitulates the time points of differentiation (Supplemental Fig. 4k-n), even though cells within a time point are heterogeneous regarding cell lineage. Branch points identify cell states or intermediates from which cells can choose paths that are dissimilar on a global level. Interestingly, early differentiating cells, including cells that

linger in this state for a long time (especially in *Jarid2* null cells), represents such a globally distinct state.

D. [Single-cell RNA vs bulk comparison] Authors use both bulk and single-cell RNA-seq to study the effect of the loss of PRC2 subunit on differentiation. To better understand the bulk RNA-seq datasets, it would be beneficial to know how heterogeneous are the cells in the intermediate time point during differentiation.

>>Response: In our experiments, we have used single cell RNA sequencing and bulk RNA sequencing to assess the intercellular differences within embryoid bodies. We have used directed differentiation to assess the ability of cells to differentiate towards mesoderm and endoderm. We have not performed single cell RNA sequencing in these directed differentiation experiments. Based on Gene Ontology analyses, however, we estimate that the cells are fairly homogenous regarding germ layer cell types (Fig. 6e – f), but we have not quantified this. The only exception is a gene ontology term of axon development (in addition to negative regulation of nervous system) in our mesoderm differentiation protocol.

E. Authors show the effect of MTF2 loss on differentiation (enhanced and faster differentiation towards all fates) though scRNA-seq and study the effect on endo- and mesodermal fates through monolayer differentiation. It would be interesting to explore the mechanism behind gene upregulation at 48h (cluster 6 in Fig6c and d): are these genes PRC2 targets in ESCs or do the authors think they could be targets of the transcription factors identified in Fig5?

>>Response: We thank the reviewer for raising a very interesting question. We have not exhaustively examined this, but we have performed new ChIP experiments for H3K27me3 and H3K4me3 marks during directed differentiation at 48hrs of mesoderm induction. We analyzed the levels of H3K27me3 on key transcription factors such as *Eomes* and *Gata6* using qPCR and found that with the loss of *Mtf2*, the levels of repressive H3K27me3 mark was further reduced during differentiation compared to the undifferentiated state (new Fig6 g-h, new Supplemental Fig 9). *Gata6* is also one of the factors upregulated during *Mtf2* loss and thus may be causally linked to the gene up-regulation we see at 48hrs of differentiation. However, we also think that it is a combination of other signaling mechanisms that were de-regulated at the undifferentiated state that might contribute to the overall gene up-regulation in the mesoderm fate (Fig. 5).

F. It is not clear how the absence of *Jarid2* causes cells to differentiate towards early precursors and neuroectodermal fates. Could the marker genes that are upregulated in the single-cell cluster of early differentiating intermediates targeted by PRC2?

>>Response: As discussed above (see reviewer 1 point 4), all three PRC2 mutants produce more neuroectodermal cells relative to wild type cells, so this effect is not specific to *Jarid2* null cells. An example of an up-regulated marker of the early differentiating intermediates cluster is *Otx2*, which is a PRC2 target gene (shown in Fig. 1a).

G. Please make the single-cell datasets explorable in a Cellxgene or Single-Cell Portal session (or equivalent tools). Computational analyses should be more thoroughly described and code made available on Github.

>>Response: We have made the single-cell dataset explorable through a shiny app called iSEE. For further instructions, we have set up a Github page to allow readers to explore the dataset through minimal steps of installation in R. The link is now added to the revised manuscript at line 469.

Other points:

- [Pag 1] Authors should provide the number of cells per genetic background/ time point, and describe the methods utilized for batch and sequencing depth normalization

>>Response: We have added in the number of cells after filtering for each genetic background in line 440 of the revised manuscript's methods section and have described the batch normalization technique in the same section. Additionally, we have provided a metadata table in the github repository for a deeper exploration of different cells from different time points and genetic background. See also Supplemental Table 1.

- [Fig1B] Scale bar is missing

>>Response: Added in scale bar of 200um in revised manuscript Fig. 1b

- [Fig1c and e] Clusters in close proximity should have distinguishable colors

>>Response: Revised Supplemental Fig. 2 provides a split view of the UMAPs according to genetic backgrounds and should provide better discerning between overlapped clusters and cells in the merged UMAP.

- [Pag5, top] "Eed null cells [...] are quite different from wild type ES cells. Within differentiating EBs, however, some of these cells acquire transcriptomes that are similar to cells with wild type or other genotypes, as observed in clusters 19 and 20 for example (Fig. 1c-d, Extended Data Fig. 1b)" Is the knockout fully penetrant or could these cells be wild-type?

>>Response: The mutant cells that we used have been described and tested for their full KO genotype by western blots and qPCR. The picture that emerges is that PRC2 mutations shift the gene regulatory balance, without strictly precluding differentiation into specific lineages. This is also apparent in the differences between EBs (where cells commit to a variety of lineages based on autocrine / juxtacrine signaling and inherent biases) and directed differentiation, where they are forced in one specific direction.

- [Pag7] The faster rate of differentiation in Mtf2 null cells is only evident in day 7.

Response: The reviewer is correct that that the progression speed analysis of Fig. 3 identifies day 4-7 (not just day 7) as the time window of faster differentiation. It is possible that this is a reflection of the analysis method (pseudotime as a function of real time), which is relatively insensitive to small changes in gene expression. In fact, we show that numerous lineage-specific genes are misexpressed in the undifferentiated stage already.

We should mention that, despite the strong differences observed between *Eed* null and the other cells, which originally led us to focus on the differences between *Mtf2* and *Jarid2*, we now have performed a trajectory analysis with *Eed* null cells included. The trajectory analysis seems to be quite robust: the new analysis (shown in Supplemental Fig. 5) recapitulates the differences between WT, *Jarid2* and *Mtf2*. Moreover, it shows increased kinetics of the *Eed* null cells that extends further into differentiation (both day 4-7 and 7-10).

- [Fig4c] Color bar is missing

>>Response: Normalized RPKM color bar scale added.

- [Fig4d and 5b] Provide number of genes in each group in figure legend or plot.

>>Response: Added in number of genes for each group in Fig. 4d GO plot and Fig. 5b

- T/ Brachyury is called differently in Fig 3e and 6b.

>>Response: Changed Brachyury into T (Fig.6b)

- Quantify statements such as “significant number”, “large part of the variance” (pag 8) or “relatively strong” (pag 9)

>>Response: Added quantitative information (numbers) in revised manuscript line 206 to elaborate on the “significant number”. For the “large part of variance” we are referring to the distance between the mentioned samples as depicted in the PCA shown in Supplemental Fig 6b (now). The “relatively strong” segment is where we used the heatmap in Fig. 4c to illustrate the differences between EZH2 signal in *Mtf2* null, *Jarid2* null and WT cells. Quantitation of these effects can be found in the box plots of Fig. 5b and Supplemental Fig. 7 (see also Perino et al 2019 Nature Genetics).

- Clarify in the figure panel that “Up-regulated genes” in Fig 5b are the 284 PRC2 targets and MTF2 upregulated from Fig4b

>>Response: Clarified by adding a panel label in Fig5b to illustrate this point

- [Fig4a, 6c-d] Provide number of genes in heatmap in figure legend or plot.

>>Response: The number of genes are included in the figure panel in the revised manuscript figures

- [Ext Data Fig1] What was the threshold to call a cluster single or mixed background in (b)? Please also add label to color bar in c-f

Response: A mixed genotype cluster is defined as a cluster where at most 80% of the cells share the same genotype (cf. Supplemental Table 1). We have included this in the legend of the figure panel

- [Ext Data Fig2d-e] Add y axis labels

>>Response: We have added y-axis labels: “Cells(%)” (Now Supplemental Figure 3)

- [Ext Data Fig3] There is a typo in the title (confirmed)
 - >>Response: Thank you. We have made the change (now in revised Supplemental Fig. 4)
- [Ext Data Fig3j-l] What is the difference between these panels?
 - >> Response: The difference between them is the time points, now correctly labelled in Supplemental Fig. 4
- [Ext Data Fig6a] Second row, last panel – x axis is not horizontal
 - >>Response: We have changed the panel accordingly

REVIEWER COMMENTS

Reviewer #1 (Remarks to the Author):

To Authors:

1. There are still major issues with the scRNA-Seq. As mentioned previously, the authors have not used the correct isogenically matched WT ESC lines for their scRNA-seq analyses. However, the revised manuscript unfortunately still has a lot of the major conclusions drawn from this flawed analyses (See Fig 2A – over/under-represented cells from each lineage in each mutant cell line). See also Figure S1A-B. This shows that the underlying transcriptional profiles of these WT cell lines are actually quite different, especially EED WT/KO versus MTF2 and J2 WT/KO. The fact that there is only one replicate of the EED WT/KO RNA-Seq is also a concern and not acceptable for publication in Nature Communications.

a. There is also the technical point that in Figure S1K in that Nanog is not down-regulated in the EED KO (just one replicate presented) and Jarid2 KO embryoid bodies. This concerns me and does not fill me with confidence when interpreting the rest of the RNA-seq results.

b. Another technical point is that the authors only selected a few example genes in Figure S1E-K to confirm expression of lineage genes during differentiation – a more systematic analysis of all meso-, endo-, and ectodermal genes is absolutely necessary.

2. The authors did not address this comment from my previous review: “Furthermore, the loss of a core PRC2 component and an accessory component during differentiation is a very important comparison which the authors make in Figures 1 and 2. Is the loss of PRC2 targeting as critical as the loss of the whole PRC2 complex? The authors include this critical control in Figures 1 and 2, but leave it out for Figures 3-6. This should be addressed by including Eed ESCs in the analyses, in particular for Figures 3 and 4”. This was a major comment and is an especially important point as the authors themselves highlight that the behavior of EED KOs is different to MTF2/J2 KOs. This is a key conceptual point that the authors did not address, as requested.

3. I felt that some novelty in the paper might be ascribed to the fact that they see certain TFs regulate the exit from pluripotency by activating the genes that lose H3K27me3. The authors show this by looking at TF motif activity in ESCs (Fig 5C). This is potentially interesting, and I did ask them to “check the occupancy of these TFs at the genes that lose H3K27me3” (to see if there was direct regulation). I felt like this would be an avenue for the authors to get at a potential mechanism behind why MTF2 KO cells are differentiating faster (outside of the loss of H3K27me3). Elucidating a specific group of TFs that specifically drive differentiation in MTF2 KOs vs JARID2 KOs would be very interesting, however, they did not do this. Instead, they ChIP’ed the levels of H3K27me3 at the promoters of the TFs, which just shows they are PcG targets.

4. A minor point is that the data presented in Fig 6G-H and all of Figure S9 have no error bars.

Reviewer #2 (Remarks to the Author):

In this considerably revised manuscript, the authors addressed most of the comments raised by all the reviewers. There is one rather minor point which the authors did not provide, and this is a metagene plot of the PRC2 targets. Instead, they argue that “a box-plot visualizes the same information (Figure 5B)”. But this is not true. Metagene profiles provide much more than a box-plot because they give a flavor of the quality of the data. The box-plots are helpful, but at least one metagene profile should be provided so that readers can assess the quality of the signal, and the level of noise.

Reviewer #3 (Remarks to the Author):

The reviewed manuscript by Loh et al now includes several new experiments and datasets that indeed strengthen most of the claims presented, and will be of interest for the community. However, a few points remained unaddressed and importantly the methods are still not described with sufficient detail to allow full comprehension and reproducibility.

Authors addressed point A.

To address B, authors added the number of high-quality cells from each genotype. However, they still do not specify how many cells were in each processed sample (WT or Mtf/ Jarid2/ Eed null in each one of day 0/4/7/10). If very similar numbers of cells were obtained for each collected sample (combination of time point and genotype) absolute numbers are fine (but number of cells in each sample need to be specified in figure legends), otherwise the use of absolute numbers is confusing as influenced by total number of cells in each individual sample.

For point C, Supplemental Figure 4 indeed helps the readers to better understand the pseudotime analysis. However, more details need to be added to the description of the pseudotime ordering in the materials and methods section, for example that HVGs (how many?) from Seurat were used as feature selection, which version of Monocle was used, . . . All elements needed to reproduce the analysis need to be described. On a related note, the scRNA analysis section should include the version of the package used.

Point D was not addressed – suggestion was to provide a visualization of single cell datasets split by genotype and time point (as in S2b, colored by time point) to help interpret possible heterogeneity (i.e. the presence of cells in both early and late differentiation stages) in matched bulk mRNA-seq (shown in S1). As this is however now possible through individual exploration of the datasets, the comment is resolved.

Point E was partially addressed in the updated figure 6.

Point F was addressed. The authors should further clarify in discussion (as done in response to point F and to reviewer 1 as well) that the Jarid1 null cells are not specifically producing more neuroectodermal cells (in the initial section of the discussion Jarid2 null cells are described as producing “more (neuro-)ectodermal” cells.)

Point G included the suggestion to make the datasets available (which has been addressed) but also to improve the description of the computational analyses (not addressed yet, see previous points). Several aspects of the analysis (see also point C, and comments below) are still unclear or confusing (which region was plotted in the heatmap in 4c generated? which TF motif database was used in 5c? which gene annotation was used for the differential expression analysis? ...) and will benefit from an expanded description in the Materials and methods section.

Additional points:

1. The number of PRC targets within the black circle in figure 4b, top and bottom, does not match (2678 vs 2679). Please revise.
2. Which genes are included in Fig 4c? From the figure legend text, it seems to include both 242 Mtf2 up-regulated genes that are PRC targets and the 58 Jarid2 up-regulated genes that are PRC targets, but the label on the y axis only mentions the 242 Mtf up-regulated genes. Moreover, the heatmap is described in figure legend as representing TSS +/- 400bp, but is described as TSS +/- 5kb in the response to reviewer 1. Please correct.
3. How were the genes in Fig 5a selected? Are they among the upregulated genes in Fig4?
4. Fig 5b, y axis is: “ChIP peaks, RPKM” – which peaks were selected? How? Description is missing in Methods section.

5. Fig 5d-I : heatmaps are described as RPKM at promoter, but how was the promoter region defined?
6. In S9b and c, figure legend clarify which genotype is shown as "mutant".

Pont-by-point response to reviewers

Response indicated as >>Response

Reviewer #1

To Authors:

1. There are still major issues with the scRNA-Seq. As mentioned previously, the authors have not used the correct isogenically matched WT ESC lines for their scRNA-seq analyses. However, the revised manuscript unfortunately still has a lot of the major conclusions drawn from this flawed analyses (See Fig 2A – over/under-represented cells from each lineage in each mutant cell line). See also Figure S1A-B. This shows that the underlying transcriptional profiles of these WT cell lines are actually quite different, especially EED WT/KO versus MTF2 and J2 WT/KO.

>> Thank you for your detailed comments. We understand your concerns and have worked very hard to address them, doing justice to the large and complex data, while also striving to present them in a lucent and comprehensible fashion.

>> We present the original single cell data with a single wild type control as face value data, which we then carefully interpret using all wild type controls in a bulk RNA-sequencing analysis. The wild type controls show some differences (which we specifically looked for and highlighted in Suppl. Fig. 1), but we are taking those in account in our interpretation and analysis. To account for differences between wild type cells, we analyzed the fold changes between the mutants and their respective wild type cells and presented this in relation to and alongside our single-cell data (Supplemental Fig. 1b-d). For this revision, we further extended the analyses of the bulk RNA-seq data (New Fig 2a, New Supplemental Fig. 5i). New Fig. 2a accompanies our analysis of cell lineage bias. With the new Supplemental Fig. 5i we compare the dynamics of marker gene expression to the dynamics observed in the single cell trajectories (speed and expression). The most relevant findings in our single cell data set, for example enhanced mesodermal differentiation in *Mtf2* null cells, strong primordial germ cell (PGC) differentiation in *Eed* null cells, and delayed differentiation in *Jarid2* null cells, are fully corroborated in the bulk RNA-sequencing data set with all isogenic controls. The ectodermal bias of *Jarid2* null cells is less clear in the bulk data, but this could also be obscured by the delayed differentiation dynamics. We also noted enhanced PGC marker expression in *Jarid2* null

cells in both the bulk and single cell data. We have restructured the text to compare these data sets in the major sections of the paper.

The fact that there is only one replicate of the EED WT/KO RNA-Seq is also a concern and not acceptable for publication in Nature Communications.

>> In the previous revision, two replicates (out of 48 in total: 6 lines, 4 time points, 2 replicates) were missing because of failed QC. We have fixed this shortcoming in this revision. The additional replicates for undifferentiated *Eed* null and WT cells look good and are now included in the analyses shown in Supplemental Fig 1 and new panel Fig 2a.

a. There is also the technical point that in Figure S1K in that *Nanog* is not down-regulated in the EED KO (just one replicate presented) and *Jarid2* KO embryoid bodies. This concerns me and does not fill me with confidence when interpreting the rest of the RNA-seq results.

>> As we and others have shown, *Eed* cells have a strong gene expression signature related to primordial germ cells (Supplemental Figs. 1c-d, 3f), which are *Nanog*-positive. Therefore, this is an expected and bona fide result. For *Jarid2* cells, the downregulation of *Nanog* expression is mainly delayed (Supplemental Fig. 5i), in line with our analysis of differentiation kinetics (Fig. 3).

>> Please note that we removed Supplemental Figure 1k in favor of the new and more extensive Supplemental Figure 5i, which also includes *Nanog* as one of the marker genes. The new location of this data is also more consistent with the structure of the Results section.

b. Another technical point is that the authors only selected a few example genes in Figure S1E-K to confirm expression of lineage genes during differentiation – a more systematic analysis of all meso-, endo-, and ectodermal genes is absolutely necessary.

>> In the previous revision we included a Gene Ontology analysis (Supplemental Figs. 1c-d), which represents an unsupervised, systematic analysis of gene expression changes. For this revision we have expanded our marker gene analysis (new Fig. 2a, new Supplemental Figure 5i). These show additional germ layer differentiation genes across the different mutants.

2. The authors did not address this comment from my previous review: “Furthermore, the loss of a core PRC2 component and an accessory component during differentiation is a very important comparison which the authors make in Figures 1 and 2. Is the loss of PRC2 targeting as critical as the loss of the whole PRC2 complex? The authors include this critical control in Figures 1 and 2, but leave it out for Figures 3-6. This should be addressed by including *Eed* ESCs in the analyses, in particular for Figures 3 and 4”. This was a major comment and is an especially

important point as the authors themselves highlight that the behavior of EED KOs is different to MTF2/J2 KOs. This is a key conceptual point that the authors did not address, as requested.

>> As we have shown, and have elaborated in the paper, the differences in *Eed* cells are more dramatic. In the revised manuscript we analyze these differences and we have commented on it. For example, we concluded: “The undifferentiated *Eed* null cells show a large number of strongly upregulated genes, which tend to be less upregulated in *Jarid2* null cells, whereas *Mtf2* null cells show relatively strong deregulation of a partially overlapping set of genes” (p.5, line 126-128).

Also, for the trajectory analysis of Fig. 3, we included a kinetics analysis with *Eed* null cells included (Suppl. Fig. 5). This trajectory of *Eed* null cells includes cells that ‘escape’ the predominant PGC fate for these cells, as they also contribute to ectodermal and mesodermal fates (clusters 13, 16, 19, 20; Supplemental Fig. 3c-d). The trajectory allows a comparison of differentiation kinetics, showing that that *Eed* null cells exhibit differentiation kinetics that appear to be a composite of those of *Mtf2* and *Jarid2* null cells (Supplemental Fig. 5).

3. I felt that some novelty in the paper might be ascribed to the fact that they see certain TFs regulate the exit from pluripotency by activating the genes that lose H3K27me3. The authors show this by looking at TF motif activity in ESCs (Fig 5C). This is potentially interesting, and I did ask them to “check the occupancy of these TFs at the genes that lose H3K27me3” (to see if there was direct regulation). I felt like this would be an avenue for the authors to get at a potential mechanism behind why MTF2 KO cells are differentiating faster (outside of the loss of H3K27me3). Elucidating a specific group of TFs that specifically drive differentiation in MTF2 KOs vs JARID2 KOs would be very interesting, however, they did not do this. Instead, they ChIP’ed the levels of H3K27me3 at the promoters of the TFs, which just shows they are PcG targets.

>> We thank the reviewer for being particularly interested in our findings that lineage-specific transcription factors are derepressed in Polycomb mutants, and that this derepression corresponds with the motifs found in other derepressed genes. We have since performed a ChIP experiment of GATA2 (which binds one of our strongly predicted motifs) on undifferentiated WT and *Mtf2* null cells, followed by quantitative assessment (RT-qPCR) of several lineage genes including *Irx3*, *Wnt7b*, *Eomes* and *Pdgfra*, to show that the loss of *Mtf2* resulted in an increase in GATA2 occupancy on regulatory elements of the *Wnt7b* and *Irx3* genes. The new data is presented and described in updated Fig. 5g (lines 301 – 306). These results point towards a feedforward loop that combines MTF2, GATA2 and several differentiation factors in a concerted effort to regulate exit of pluripotency. We have also made changes to our model in Fig.7 to elaborate this mechanism more in detail.

4. A minor point is that the data presented in Fig 6G-H and all of Figure S9 have no error bars.

>> We have included all the data points of the replicates done for the qPCR experiments in the figures mentioned.

Reviewer #2 (Remarks to the Author):

In this considerably revised manuscript, the authors addressed most of the comments raised by all the reviewers. There is one rather minor point which the authors did not provide, and this is a metagene-plot of the PRC2 targets. Instead, they argue that “a box-plot visualizes the same information (Figure 5B)”. But this is not true. Metagene profiles provide much more than a box-plot because they give a flavor of the quality of the data. The box-plots are helpful, but at least one metagene profile should be provided so that readers can assess the quality of the signal, and the level of noise.

>> Thank you for your support of our work. Overall we observe very good signal to noise ratios, and we have provide a metagene plot for our analyses in Figure 5b in Supplemental Fig 7b.

Reviewer #3 (Remarks to the Author):

The reviewed manuscript by Loh et al now includes several new experiments and datasets that indeed strengthen most of the claims presented, and will be of interest for the community.

>> Thank you for your support of our work.

However, a few points remained unaddressed and importantly the methods are still not described with sufficient detail to allow full comprehension and reproducibility.

Authors addressed point A.

To address B, authors added the number of high-quality cells from each genotype. However, they still do not specify how many cells were in each processed sample (WT or Mtf/ Jarid2/ Eed null in each one of day 0/4/7/10). If very similar numbers of cells were obtained for each collected sample (combination of time point and genotype) absolute numbers are fine (but number of cells in each sample need to be specified in figure legends), otherwise the use of absolute numbers is confusing as influenced by total number of cells in each individual sample.

>> We have provided an extra table in Supplemental Table 1 to specify how many cells we have for each timepoint and each genetic background.

For point C, Supplemental Figure 4 indeed helps the readers to better understand the pseudotime analysis. However, more details need to be added to the description of the pseudotime ordering in the materials and methods section, for example that HVGs (how many?) from Seurat were used as feature selection, which version of Monocle was used, ... All elements needed to

reproduce the analysis need to be described. On a related note, the scRNA analysis section should include the version of the package used.

>> We have expanded our methods section to now include elements mentioned by the Reviewer in lines 462 to 472.

Point D was not addressed – suggestion was to provide a visualization of single cell datasets split by genotype and time point (as in S2b, colored by time point) to help interpret possible heterogeneity (i.e. the presence of cells in both early and late differentiation stages) in matched bulk mRNA-seq (shown in S1). As this is however now possible through individual exploration of the datasets, the comment is resolved.

Point E was partially addressed in the updated figure 6.

Point F was addressed. The authors should further clarify in discussion (as done in response to point F and to reviewer 1 as well) that the *Jarid1* null cells are not specifically producing more neuroectodermal cells (in the initial section of the discussion *Jarid2* null cells are described as producing “more (neuro-)ectodermal” cells.)

>> We thank the reviewer for the point and have since clarified our *Jarid2* findings clearer in the discussion (line 367).

Point G included the suggestion to make the datasets available (which has been addressed) but also to improve the description of the computational analyses (not addressed yet, see previous points). Several aspects of the analysis (see also point C, and comments below) are still unclear or confusing (which region was plotted in the heatmap in 4c generated? which TF motif database was used in 5c? which gene annotation was used for the differential expression analysis? ...) and will benefit from an expanded description in the Materials and methods section.

>> We thank the reviewer for pointing out some of these missing elements. We have since included a legend in Fig 4c to explain the genomic region of +/- 0.4kb that we used for the heatmap and added the necessary information such as TF motif database to the methods section.

Additional points:

1. The number of PRC targets within the black circle in figure 4b, top and bottom, does not match (2678 vs 2679). Please revise.

>> We have fixed this error in the new Fig4b

2. Which genes are included in Fig 4c? From the figure legend text, it seems to include both 242 *Mtf2* up-regulated genes that are PRC targets and the 58 *Jarid2* up-regulated genes that are PRC targets, but the label on the y axis only mentions the 242 *Mtf* up-regulated genes. Moreover, the

heatmap is described in figure legend as representing TSS +/- 400bp, but is described as TSS +/- 5kb in the response to reviewer 1. Please correct.

>> We have revised the figure to depict the correct TSS window of +/- 400bp

3. How were the genes in Fig 5a selected? Are they among the upregulated genes in Fig4?

>> Genes in Fig5a were selected among the up-regulated genes in *Mtf2* null cells during undifferentiated state.

4. Fig 5b, y axis is: “ChIP peaks, RPKM” – which peaks were selected? How? Description is missing in Methods section.

>> We have included a description of this in our Methods section lines 493-495)

5. Fig 5d-I : heatmaps are described as RPKM at promoter, but how was the promoter region defined?

>> Promoter regions are defined by a genomic window of +/- 500bp from the TSS of the genes.

6. In S9b and c, figure legend clarify which genotype is shown as “mutant”.

>> In S9b and c, the legends below each figure depicts the mutant and WT bars with or without treatment of EED226.

REVIEWERS' COMMENTS

Reviewer #1 (Remarks to the Author):

I am unfortunately not convinced by the author's arguments about the lack of matched wild-type controls in the scRNA-seq analyses in Figure 1. In my view, this is a fundamental flaw that has not been addressed adequately.

While they have performed ChIPs of GATA2 in the revised paper, the necessary controls are missing to support the main message - in particular, they did not CHIP in Jarid2 null or EED null to show that these TFs would be driving the specific transcriptional program in the MTF2 null cells (and hence why the lineage choices are different in the different KOs).

Reviewer #3 (Remarks to the Author):

The Authors extensively revised the manuscript and addressed the points raised by all Reviewers. A very minor last aspect to be addressed is the absence of a description of the bandplot/metagene plot provided in Suppl. Fig. 7b following the comment from Reviewer 2 - what is shown in black and what percentage of the data are the darker and lighter colors representing? A description should be added in Methods or Suppl. Fig. legend.

Pont-by-point response to reviewers

Response indicated as >>Response

Reviewer #1

I am unfortunately not convinced by the author's arguments about the lack of matched wild-type controls in the scRNA-seq analyses in Figure 1. In my view, this is a fundamental flaw that has not been addressed adequately.

While they have performed ChIPs of GATA2 in the revised paper, the necessary controls are missing to support the main message - in particular, they did not CHIP in Jarid2 null or EED null to show that these TFs would be driving the specific transcriptional program in the MTF2 null cells (and hence why the lineage choices are different in the different KOs).

>> In multiple revisions we have attempted to satisfy the reviewer. We believe that the incorporation of our bulk-RNA seq analyses with matched WT cells substantiates our findings. The ChIP experiment is not really missing controls as the reviewer is suggesting (the matched WT cells are included), the reviewer is rather suggesting a different experiment than the one we did (ChIP in Jarid2 and Eed null cells). Our experiment uncovers a MTF2-GATA2 feedforward loop, the absence of which simultaneously reduces repression by PRC2 and stimulates lineage-specific transcription networks and endogenous signaling.

Reviewer #2 (Remarks to the Author):

The Authors extensively revised the manuscript and addressed the points raised by all Reviewers. A very minor last aspect to be addressed is the absence of a description of the bandplot/metagene plot provided in Suppl. Fig. 7b following the comment from Reviewer 2 - what is shown in black and what percentage of the data are the darker and lighter colors representing? A description should be added in Methods or Suppl. Fig. legend.

>> We thank Reviewer 2 for the comments. We have since included a description in the Figure legend of the metagene plot for Suppl. Fig. 7b to explain the following - "The median enrichment is visualized as a black line with the 50th and 90th percentile as a dark and light colour respectively."